# HARD-KV: Head-Adaptive Regularization for Decoding-time KV Compression

Yuxuan Yang [1]   Feiyang Ren [1]   Bowen Zeng [1]   Dalin Zhang [2]   Jinpeng Chen [3]   Gang Chen [1]   Huan Li [1]

## Abstract

Long-context LLM inference faces a fundamental conflict: head-adaptive compression algorithms (e.g., Top-$p$ nucleus sampling) offer superior accuracy by dynamically fluctuating memory budgets, yet modern inference engines (e.g., vLLM) demand rigid, static memory patterns to leverage CUDA Graphs and PagedAttention. We resolve this "Static-Dynamic" mismatch with HARD-KV, a unified framework that that bridges dynamic selection with rigid system constraints. HARD-KV introduces a Cascade Cache hierarchy, managing the token lifecycle across dense, sparse, and condensed tiers. Crucially, we propose a Logits Calibration mechanism that normalizes diverse importance metrics into a unified probability space, enabling consistent Top-$p$ budgeting across heterogeneous heads. To bridge the efficiency gap, we offer a system-level solution, which rewrites fragmented, dynamic indices into contiguous physical layouts compatible with high-performance inference engine. Extensive experiments on math-reasoning benchmarks (AIME, U-Math) verify that HARD-KV achieves up to $2\times$ throughput improvement over static baselines while maintaining high-fidelity generation in 10k+ token scenarios. Code is available at https://github.com/SuDIS-ZJU/HARDInfer.

## 1. Introduction

The deployment of Large Language Models (LLMs) in long-context scenarios is increasingly constrained by the linear growth of Key-Value (KV) cache memory. As sequence lengths extend into the tens of thousands, the memory footprint of the KV cache frequently exceeds that of the model weights, creating a critical bottleneck for throughput and latency. To mitigate this, decoding-time compression methods (Liao et al., 2025; Wu et al., 2025; Behnam et al., 2025; Jo et al., 2025; Zeng et al., 2025; Zhang et al., 2025a) have emerged, dynamically reducing memory usage by retaining only the most critical tokens during generation.

A central tension in this domain lies between algorithmic accuracy and system efficiency. Recent research (Fu et al., 2024; Xiao et al., 2024; Tang et al., 2024) indicates that "head-adaptive" strategies—which allow cache size to fluctuate based on the specific information density of each attention head—offer superior performance. This accuracy is crucial for extending compression to decoding time, particularly for tasks like mathematical reasoning where errors accumulate over long generation trajectories (Chen et al., 2025a). Lin et al. (2025) argue that the superiority of head-adaptive budget allocation stems from the intrinsic dynamism of attention, demonstrating that simple-yet-effective Top-$p$ (nucleus sampling (Holtzman et al., 2019)) is sufficient for selection. Similarly, other works (Xuanfan Ni et al., 2025; Feng et al., 2025b;a) propose new criteria for performance-prioritized compression that rely on dynamic allocation.

However, this dynamic nature is fundamentally incompatible with modern inference engines. The uneven, unpredictable, and fragmented memory indices generated by head-adaptive selection conflict with standard optimizations such as PagedAttention (Kwon et al., 2023a), continuous batching (Yu et al., 2022), and CUDA Graphs (NVIDIA Corporation, 2024), all of which rely on *static* or *regularized* memory block structures.

> **Challenge:** We characterize the core conflict as the gap between *Dynamic, Context-Dependent Attention Patterns* and *Static, Request-Independent System Designs*.

To bridge this gap, we propose HARD-KV (**H**ead-**A**daptive **R**egularization for **D**ecoding-time **KV** Compression), a framework designed to enable dynamic cache allocation within the constraints of modern inference architectures.

To bridge this gap, HARD-KV introduces a Cascade Cache architecture that organizes KV memory into three sequence-level tiers. We further propose a Logits Calibration procedure that converts Top-$k$ selections into Top-$p$-aligned coun-

---

[1]The State Key Laboratory of Blockchain and Data Security, Zhejiang University [2]Space Information Research Institute, Hangzhou Dianzi University [3]School of Computer Science (National Pilot Software Engineering School), BUPT. Correspondence to: Huan Li <lihuan.cs@zju.edu.cn>.

terparts, allowing diverse KV selection strategies to share a unified head-adaptive budget allocation mechanism. This calibration produces cache patterns that better follow the raw attention distribution, making them naturally compatible with the three-tier cache. Finally, HARD-KV regularizes dynamic head-wise indices through a customized inference system that combines memory rewriting and sparse loading. This design reconciles adaptive token selection with the structural constraints of modern inference engines, preserving algorithmic accuracy while improving system efficiency. On math-reasoning benchmarks, including AIME and U-MATH, HARD-KV achieves up to a $1.5\times$ performance gain and a $2\times$ throughput improvement.

To summarize, our work unifies head-adaptive KV cache management into a robust system that is both algorithmically performant and system-efficient We summarize our technical contributions as follows:

1. **Unified Cascade Cache Architecture:** We introduce a multi-tier memory hierarchy that standardizes the token lifecycle. This structure accommodates various KV selection methods, unifying diverse compression techniques into a single pipeline. (Section 3.1)

2. **Logits Calibration for Effective Top-$p$ Budgeting:** We propose a calibration method connecting Top-$k$ and Top-$p$ selection. By mapping distribution alterations onto the raw attention distribution, we enable Top-$p$ budgeting to dynamically adapt to attention patterns rather than relying on fixed limits. (Section 3.2)

3. **System-Aware Index Regularization:** We provide a method to integrate dynamic compression into rigid system architectures. Our regularization for head-adaptive indices resolves compatibility issues with CUDA Graphs, ensuring high system efficiency with minimal metadata overhead.(Section 3.3)

## 2. Related Work

**Adaptive KV Cache Compression**   KV cache compression (Li et al., 2024) optimizes LLM inference SLOs (e.g., latency, throughput) by reducing memory footprints during attention computation. Unlike Efficient Attention methods (Yang et al., 2024; Ainslie et al., 2023; Fu et al., 2025) designed for full-stack industrial deployment, KV cache compression has largely remained a research testbed, lacking robust integration with modern inference engines.

Recent studies (Fu et al., 2024; Feng et al., 2024; Du et al., 2025; Zeng et al., 2024) have pivoted from uniform compression to *head-adaptive strategies*, driven by the observation of attention heterogeneity. Methods like AdaKV (Feng et al., 2024) and HeadKV (Fu et al., 2024) implement head-wise allocation based on metrics like eviction loss and importance

scores. Similarly, Du et al. (2025) use policy optimization to mix full and sparse attention, showing that heterogeneous configurations outperform uniform baselines.

Concurrently, approaches like Twilight (Lin et al., 2025) and MagicPig (Chen et al., 2024b) have shown that Top-$p$ (cumulative probability) selection yields higher fidelity than rigid Top-$k$ (fixed count) truncation. Despite the superiority of Top-$p$ selection in ideal scenarios, robustly integrating these dynamic strategies remains a challenge: existing methods often decouple sampling from memory management, while diverse importance metrics (e.g., L2 norm vs. attention scores) defy unified probabilistic budgeting.

HARD-KV bridges this gap with a *logits calibration* mechanism that normalizes diverse selection metrics into a unified probability space, enabling consistent Top-$p$ budgeting across heterogeneous heads while enforcing strict, system-compatible regularity on the resulting memory layout.

**System-Algorithm Co-Design in LLM Inference**   State-of-the-art inference engines maximize throughput by enforcing rigid, predictable execution patterns. Innovations such as RadixAttention (Zheng et al., 2024), Tree-Attention (Cai et al., 2024), and static load-balancing (Liu et al., 2024) demonstrate the efficacy of co-designing algorithms with static system constraints.

Likewise, deploying head-adaptive KV compression introduces an algorithm-system conflict: the memory allocations *vary* at every step, violating the static assumptions of engines built for regular allocation. Although head-wise sparse attention is computationally feasible via kernel support like FlashInfer (Ye et al., 2025), we identify three systemic barriers that prevent theoretical compression rates from actual speedups:

*1) CUDA Graphs Incompatibility:* CUDA Graphs rely on static tensor shapes and persistent pointers. The variable cache sizes produced by Top-$p$ selection force the system to either frequently recapture graphs or revert to eager execution, neutralizing latency gains.

*2) PagedAttention Fragmentation:* Current paging mechanisms in real-time LLM serving (Yu et al., 2022; Kwon et al., 2023a) assume uniform memory distribution. Head-adaptive eviction shatters this uniformity, causing severe fragmentation that breaks the block contiguity required for efficient virtual-to-physical mapping.

*3) KV Selection Overhead:* Standard KV selection requires computing $\mathcal{O}(n^2)$ attention maps, which is incompatible with FlashAttention (Dao, 2023). Such a cost introduces significant decoding overhead, stalling token generation.

Unlike prefill-phase compression (Li et al., 2024; Chen et al., 2024a), which occurs only once, our work targets

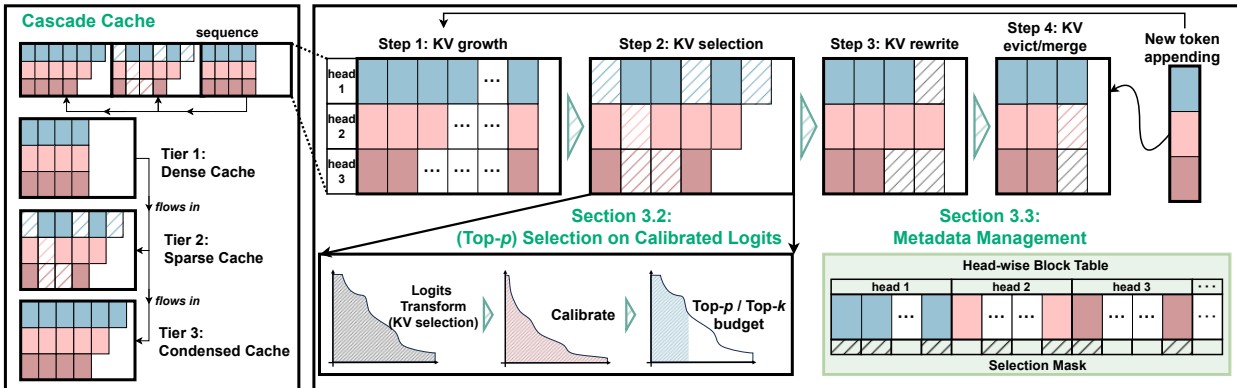

*Figure 1.* **Left**: The *Cascade Cache* hierarchy manages the token lifecycle across three storage tiers: Dense (recent), Sparse (adaptive), and Condensed (archival). **Right**: The HARD-KV execution pipeline. (Step 1) The KV cache grow in context; (Step 2) When compression is triggered, different selection methods will be calibrated towards real attention distribution and conduct dynamic Top-$p$ sampling; (Step 3) Upon the sparse mask generated in the previous step, the KV indices will be condensed by rewriting; (Step 4) The KV cache can be further reduces to satisfy hard constraints by eviction or merging.

periodic compression during the latency-critical *decoding* phase. Proceeding to existing works (Behnam et al., 2025; Wu et al., 2025; Jo et al., 2025), we consider it crucial to maintain CUDA Graphs compatibility over tens of thousands of decoding steps in long-context generation. We address this by integrating head-adaptive compression into a custom engine built on NANO-VLLM (Yu Xingkai, 2025). While prior work, such as DiffKV (Zhang et al., 2025b), manages dynamic head-wise memory allocation via bidirectional page tables and GPU-side circular lists, we take a different approach. We focus on algorithm-system integration, proposing a novel hybrid of sparse and condensed formats to systematically tame head adaptivity.

## 3. The HARD-KV Design

### 3.1. The Framework Overview

**The Cascade Cache Hierarchy**   Standard self-attention suffers from linear memory complexity $\mathcal{O}(n)$. While prior works have explored either *eviction* (maintaining a fixed budget) or *sparse retrieval* (dynamic loading), these approaches often treat KV memory as a monolithic structure. We propose **Cascade Cache**, a hierarchical architecture that unifies these paradigms by managing the KV lifecycle across three distinct tiers (Figure 1, Left):

- **Tier 1: Dense Cache.** Buffers the most recent tokens in contiguous memory. This ensures full-fidelity attention for the local window, preserving "attention sinks" (Xiao et al., 2023) and immediate syntactic dependencies.

- **Tier 2: Sparse Cache.** As tokens age out of the dense window, they transition to the Sparse Cache. We apply *Calibrated Top-p Selection* (Section 3.2) to retain

only information-dense pairs, effectively implementing head-adaptive budgeting based on attention entropy.

- **Tier 3: Condensed Cache.** To enforce hard memory limits, the oldest blocks are further compressed into constant-sized storage via eviction or weighted merging.

**The Continuous Compression Pipeline**   Building upon the Cascade Cache architecture, HARD-KV orchestrates the KV cache lifecycle via an asynchronous four-step pipeline (Figure 1, Right). We implement this within a custom NANO-VLLM engine (Yu Xingkai, 2025) to ensure strict compatibility with *continuous batching* and *CUDA Graphs* (see details in Appendix A). **Step 1**: The decoding requests linearly append KV pairs to the Tier 1 Dense Cache, utilizing efficient append-only kernels. **Step 2**: When the dense window reaches a predefined threshold, the *Logits Calibration* mechanism activates. It normalizes diverse importance metrics into a unified probability space, generating a head-adaptive sparse mask via Top-$p$ sampling (details in Section 3.2). **Step 3**: To resolve the static-dynamic mismatch, the *Index Regularization* engine rewrites the discontinuous selected tokens into a regularized, contiguous physical layout. **Step 4**: Finally, to maximize memory utilization, the oldest blocks are compacted via eviction or merging before new dense blocks are allocated.

We adapt the block table to the head-wise format with selection masks indicating effective cache. A metadata management system is coupled to support head-adaptive dynamic budget. This system is critical for enabling high-throughput CUDA Graphs execution and preventing memory fragmentation (details in Section 3.3).

This pipeline can operate asynchronously; the Dense Cache

handles immediate writes at the end of the sequence; meanwhile, compression (Steps 2-4) operates on established history, minimizing latency impact.

With the efficient Cascade Attention (see Appendix F) kernel implementation provided by Flashinfer (Ye et al., 2025), HARD-KV can compute the exact full-sequence attention from the partial attention outputs of three tiers in the proposed Cascade Cache.

**Cautious Update Strategy**  A critical refinement in the Tier 1 → Tier 2 transition is the *Cautious Update* strategy. Unlike methods that compress based on a single snapshot (e.g., the mean average attention in the last window), we treat the dense window traversal as a "voting" period. We maintain a cumulative selection mask from *all* queries in the window and only compress a block after it fully exits the dense tier. For a strict constraint on the budget, we use ranking on the votes for selecting candidates to preserve. This prevents the premature eviction of tokens that are temporarily irrelevant to the current step but critical for the local context.

### 3.2. Logits Calibration for Effective Top-$p$ Budgeting

> This section proposes a unified framework to normalize heterogeneous methods, facilitating fair evaluation and control via *a single hyper-parameter $p$*.

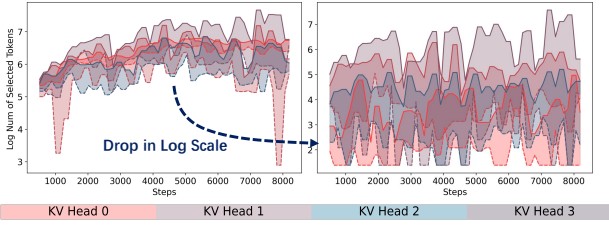

*Figure 2.* Comparison of the log number of selected tokens ($y$-axis) under P90. Solid lines ("—") represent the upper bound, while dotted lines ("- - -") represent the lower bound. **Left:** Selections based on raw logits. **Right:** Selections based on max-pooled logits (SNAPKV).

**Unified Top-P Selection**  Existing Key-Value (KV) selection methods typically encode inductive biases, such as attention locality and token redundancy, directly into attention scores. While such heuristics are effective for static Top-$k$ selection, they are often misaligned with the inherently context-dependent nature of Top-$p$ budgeting.

Ideally, a fixed probability threshold $p$ should produce a consistent retained-token budget across different selection methods. However, we observe that directly normalizing logits modified by different priors can cause their selection patterns to deviate substantially from those induced by the raw logits. We refer to this phenomenon as *selection col-*

*lapse* (Figure 2): a sharp reduction in the retained-token budget under the same probability threshold $p$. As shown in the figure, compared with the raw attention baseline, applying a selection method such as SNAPKV reduces the retained budget from roughly $1000$ ($\approx e^7$) tokens to only $10$ ($\approx e^3$) tokens under the same $p$-density at P90.

To address this issue, we first unify diverse KV selection operations into a three-step formulation, which allows most existing selection methods to be adapted to a Top-$p$ budget; details are provided in Appendix E.

We address this by first unifying diverse operations into a three-step framework capable of adapting most selection methods to a Top-$p$ budget (refer to Appendix E).

Furthermore, we propose an *order-preserving* calibration method to facilitate Top-$p$ sampling without compromising the underlying selection strategies. This approach maps distributions from methods like SNAPKV and R-KV to a normalized softmax distribution. Crucially, it preserves the $(k, p)$ mapping (i.e., linking Top-$k$ sets to cumulative probabilities $p$) by adjusting the temperature of the attention scores.

**Problem 1.** *Given a threshold $p$, let Top-$p$ sampling over $\{S_i\}$ yield $k$ tokens for a specific head. The probability mass for these $k$ attention scores is denoted by $M_k(S) = \sum_{i \in Top\text{-}k} S_i$. We aim to minimize the error between the probability mass of the raw attention scores and the shifted scores by adjusting the temperature $T$:*

$$\min_T \left\| M_k\left(S\left(\frac{1}{T}\right)\right) - M_k\left(\hat{S}\left(\frac{1}{T}\right)\right)\right\|.$$

In the standard setup, temperature scaling is defined as $S(\frac{1}{T}) = S^{\frac{1}{T}}$. However, for methods where $\hat{S}$ is not guaranteed to be a valid probability distribution, directly computing $\hat{S}^{\frac{1}{T}}$ renders the problem intractable. Therefore, we extend the definition of temperature control to operate on the altered scores (i.e., $\hat{S}$) rather than the raw logits.

Under the novel setup, we still expect to guarantee the invariance of the sampling results for the original methods.

**Constraint 1.** *Order-invariance: Let the distribution alteration on the raw logits $z$ be formalized by a function $g(z)$. We construct $\hat{S}(g(z), T)$ such that $\hat{S}|_k \equiv \hat{S}(g(z), T)|_k$, where $S|_k$ denotes the Top-$k$ subset of $S$.*

Under this definition, $\hat{S}(\cdot)$ can be any function that simplifies the solution to Problem 1 while retaining the intuitive properties of "temperature". We investigate the following two scenarios:

**Solution 1.** *If $g(z)$ preserves the order of $\hat{S}$, we may optionally define $\hat{S} = S(g(z/T))$.*

Examples of order-preserving (See Condition 1 in Appendix D.2) of $g(z)$ include max-pooling in SNAPKV and

RocketKV, key quantization in Twilight, and vanilla attention averaging in H2O.

**Solution 2.** *If $g(z)$ is arbitrary, we define $\hat{S} = S((g(z)/T)$.*

Solution 2 applies to methods such as the pairwise scoring between keys in R-KV and KeyDiff (Park et al., 2025).

In both scenarios, we identify the optimal temperature $T$ for each KV head using either binary search (zero-order optimization) or gradient descent (first-order optimization). These approaches yield a tempered attention distribution that satisfies Constraint 1 while achieving near-optimal Top-$k$ mass preservation error, denoted by $\|M_k(S) - M_k(\hat{S})\|$. The complete algorithm and associated proofs are detailed in Appendix D.

This *Calibration* enables the fair evaluation on Top-$p$ patterns for methods originally designed exclusively for Top-$k$ selection.

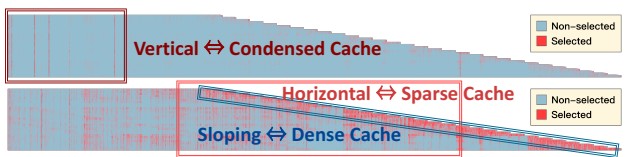

*(a)* The P60 and P90-P60 token selection heatmap from raw attention scores in layer 6 averaged on heads.

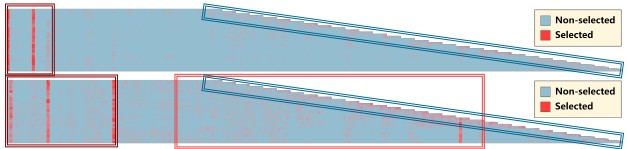

*(b)* The P60 and P90-P60 token selection heatmap from maxpooled scores (SNAPKV) in layer 6 averaged on heads.

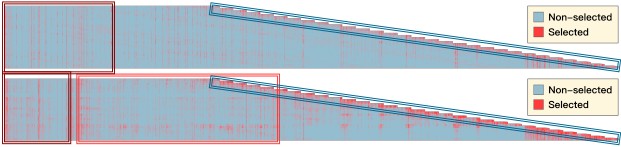

*(c)* The P60 and P90-P60 token selection heatmap from similarity enhanced scores (R-KV) in layer 6 averaged on heads.

*Figure 3.* Three different patterns observed in different resolutions of the selection heatmaps, corresponding to three tiers in Cascade Cache. The figure is generated by stacking binary attention selection maps of all KV heads, where RED represents for selected and TEAL represents for non-selected.

**Unified Patterns in Top-$p$ Sampling** Following calibration, we can precisely evaluate the effectiveness of Top-$p$ sampling on two representative methods originally designed for alternative selection criteria:

- SNAPKV: Applies max-pooling over the sequence and selects a Top-$k$ budget based on the pooled attention scores.

- R-KV: Computes the cosine similarity between keys and prunes based on a fixed similarity threshold. The Top-$k$ selection is derived from a weighted sum of attention and similarity scores.

We analyze the effectiveness of Top-$p$ sampling by partitioning the selection results into two distinct categories: **low-resolution** and **high-resolution**.

*Low-resolution* refers to the subset of tokens selected under a lower probability threshold $p$ (e.g., P60). Selection results in this regime typically concentrate on time-invariant tokens, exhibiting vertical patterns.

*High-resolution* is defined as the incremental subset of tokens required to reach a higher threshold $p$. Unlike the low-resolution set, high-resolution selection results are often dispersed and exhibit turbulent distribution as the sequence length increases, forming horizontal patterns. These patterns frequently demonstrate periodic or unpredictable retention of tokens that were not selected in recent decoding steps.

In both regimes, we observe an accumulation of attention scores within the sliding window, manifesting as sloping patterns. This observation reinforces the common practice of maintaining a local window for the most recent tokens.

With calibration, different methods can yield similar densities in the heatmaps in Figure 3 under a unified $p$ metric. By aligning the Top-$p$ budget with constraints on probability mass, we observe comparable patterns across methods, albeit with varying emphases. This alignment framework further supports the development of compression methods that preserve the fundamental properties of attention computation (Appendix G).

### 3.3. A System View of KV Index Management

> This section introduces how to unify the layer-wise varying KV indices into a *fixed buffer* storing indices for different heads in support for CUDA Graphs.

#### 3.3.1. REGULARIZATION OF HEAD-ADAPTIVE KV INDICES

The above head-adaptive Top-$p$ selection for KV Cache will incur challenges for inference systems. To ensure compatibility with CUDA Graphs, the indices and offsets for KV cache pages (or blocks) must be fixed across all layers.

To this end, we explore combinations of the following three index regularization operations:

1. *Rewrite Cache.* The most direct approach is to read the selected blocks and write them into a regularized, contiguous range.

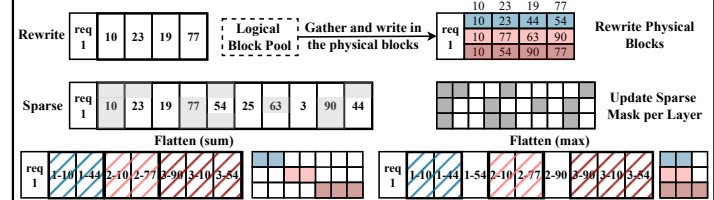

*Figure 4.* **Left:** The process of KV cache indexing in PagedAttention. The inference engine uses batch-level block table to collect indices in global-level block pool that point to physical KV blocks. Please refer to Appendix B.1 for further details. **Right:** Three techniques to regularize head-adaptive KV blocks.

2. *Sparse Loading.* Hardware-efficient attention masks are applied during kernel execution for layer-wise dynamic loading. By enabling masking, indices can be regularized at the cost of redundant KV cache loading.

3. *Flatten Index.* Indices are flattened into a 1-dimensional array. Using this technique, we can regularize per-layer block indices to either the maximum length (i.e., **Max**, yielding higher utilization of the loaded KV cache) or the sum over lengths (i.e., **Sum**, requiring fewer rewrite operations).

Based on the constraints imposed on KV cache index layouts, we categorize the design space into four tasks. Detailed configurations and corresponding experiments are provided in Appendix B.2.

We focus here on the most flexible and systematically challenging setting, **Head-wise Allocation (HA)**, and empirically compare strategies constructed from the three index regularization operations above.

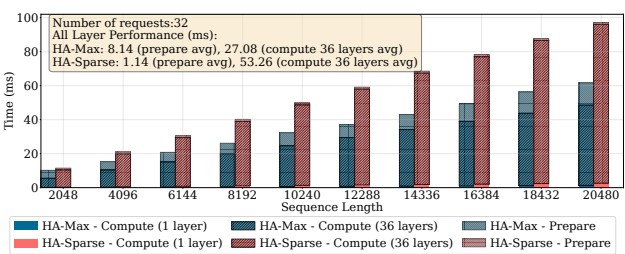

*Figure 5.* The latency evaluation for solutions in the Head-wise Allocation (HA) task. See Appendix B.2 for ablations of other tasks.

**Task: Head-wise Allocation (HA).** Different heads have different budgets, and independently selected indices.

**Solutions: HA-Sparse** - vanilla sparse loading for head-flattened indices; - rewrite KV Cache to the maximum number of allocated blocks, with often higher preparation operation and higher computation efficiency in longer context.

As shown in Figure 5, **HA-Sparse** always retains a lower cost at preparation stage and is comparably efficiency when the sequence length is limited; **HA-Max** often incurs higher

preparation latency due to the rewrite operation. However, when the sequence extends to over 10 thousand, **HA-Max** can cut the computations latency in half. The above analysis demonstrates the necessity in rewriting KV cache into condensed blocks to maximize throughput, especially in long decoding.

In our system design, we set two budget threshold of $L_{lower}$ and $L_{upper}$. When exceeding $L_{lower}$, the system will trigger selections for sparse loading; when exceeding $L_{upper}$, the system will rewrite and organize KV cache layouts in contiguous space.

### 3.3.2. PRINCIPLES FOR SMART INDEX MANAGEMENT

We present three governing principles (P1–P3) for management and regularization of sparse KV Cache metadata.

**P1: Joint optimization of memory occupancy and latency.**

Cache rewriting typically incurs a higher preparation cost than sparse loading, due to additional bandwidth pressure and memory movement during the rewrite. However, this upfront cost can be amortized by the resulting reduction in per-step decoding latency, thereby improving compute-side efficiency. We characterize this trade-off using a *Memory-Time Integral* metric, which jointly accounts for memory occupancy and execution time and thus provides a clearer view of system-level throughput. A detailed discussion is provided in Appendix B.2.

**P2: Head-wise allocation for block indices.**

A practical and efficient solution for fine-grained KV cache allocation (head-wise allocation) is to revise the global or batch-level block table metadata (See Appendix B.2). Unlike existing methods that employ index mapping across all layers and heads, our design utilizes head-independent indices for physical cache regularized for all layers. Crucially, this data structure supports CUDA Graphs, capturing kernel executions throughout the model's depth.

**P3: Sparsity-agnostic management.**

Two factors are critical in the latency breakdown for

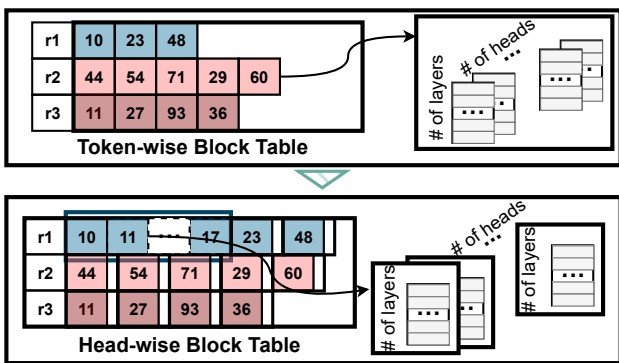

*Figure 6.* Flattened Head-wise block table.

task **HA**: overall sparsity and maximum sparsity (previously set for 12.5% and 50%, respectively). When holding overall sparsity constant, the maximum sparsity dictates the load budget required for the attention computation. Conversely, given a fixed maximum sparsity, the overall sparsity represents the total utilization within that budget. Our latency analysis indicates that maximum sparsity exerts a relatively smoother impact on overall latency compared to variations in overall sparsity.

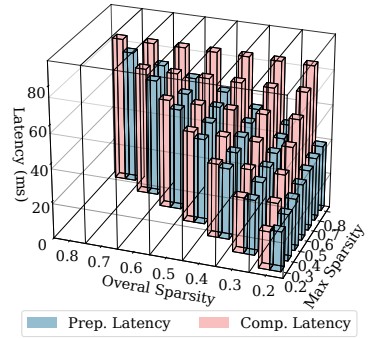

*Figure 7.* Latency breakdown by max sparsity per head and overall sparsity.

## 4. Experiment

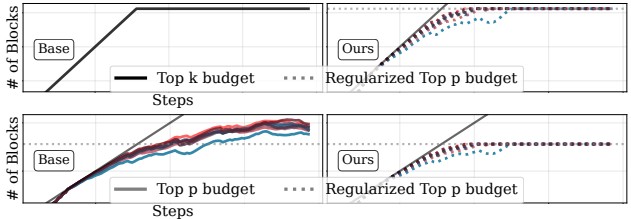

*Figure 8.* Illustrations for experiment settings. **Upper**: Fixed Top-$k$ budget. Base: always keep $k$ blocks; Ours: $k$ as the maximum block usage for Top-$p$ pruned each head. **Lower**: Dynamic Top-$p$ budget. Base: keep tokens by Top-$p$ metrics; Ours: constrain the growth of the block usage.

We evaluate our framework under two settings:

1. **Fixed (Top-$k$) Budget.** (Figure 8, Upper) We adapt selection methods to regularized Top-$p$ budgets where $k$ limits the maximum KV Cache consumption ($L_{upper}$,

*Table 1.* **Fixed Budget Performance (Top-$k$).** Comparison of baselines vs HARD-KV across budget sizes {1024, 2048, 4096}. Top-$p$ budget for our methods is 0.90. *Full Attn* represents performance without budget constraints.

| Method | Config | Budget Size | | | | | |
|--------|--------|-------------|---|---|---|---|---|
| | | **1024** | | **2048** | | **4096** | |
| | | Acc | Len | Acc | Len | Acc | Len |
| *Dataset: AIME24* | | | | | | | |
| *Full Attention (Ref.)* | | **Acc**: 79.9% | | **Len**: 13735.7 | | | |
| Vanilla | Base | 10.0% | 32161.0 | 33.1% | 23530.6 | 29.7% | 31843.5 |
| | + Ours | **24.0%** | **24158.1** | **37.3%** | **24158.1** | **33.5%** | **23449.8** |
| SnapKV | Base | 6.2% | 31323.1 | 21.3% | 23860.3 | 57.7% | 19157.4 |
| | + Ours | **13.3%** | **32768.0** | **40.2%** | **28471.4** | 56.7% | 21616.5 |
| RKV | Base | 13.3% | 30423.2 | 23.3% | 27968.0 | 43.3% | 23281.5 |
| | + Ours | **23.4%** | **30480.8** | **53.3%** | 27819.6 | **57.9%** | **28094.0** |
| *Dataset: AIME25* | | | | | | | |
| *Full Attention (Ref.)* | | **Acc**: 63.3% | | **Len**: 17528.8 | | | |
| Vanilla | Base | 6.5% | 32309.7 | 16.7% | 26617.5 | 24.8% | 31851.4 |
| | + Ours | **20.5%** | **28086.2** | 12.9% | **27929.5** | 20.7% | **28794.4** |
| SnapKV | Base | 3.1% | 31007.5 | 19.3% | 26291.5 | 27.9% | 25621.4 |
| | + Ours | **3.3%** | **31796.3** | **23.3%** | **30152.4** | **42.3%** | 25228.6 |
| RKV | Base | 10.8% | 31879.5 | 21.9% | 27706.7 | 40.0% | 23944.3 |
| | + Ours | **17.8%** | **32279.0** | **26.7%** | **31068.6** | **56.7%** | **28231.6** |
| *Dataset: UMath* | | | | | | | |
| *Full Attention (Ref.)* | | **Acc**: 50.4% | | **Len**: 9815.7 | | | |
| Vanilla | Base | 9.8% | 32768.0 | 38.0% | 16102.5 | 32.1% | 26314.3 |
| | + Ours | **46.1%** | **15363.5** | 36.2% | **16791.5** | 36.1% | **16118.8** |
| SnapKV | Base | 18.0% | 22350.7 | 41.8% | 19021.4 | 46.7% | 12420.2 |
| | + Ours | 15.5% | 27705.7 | **46.2%** | 22499.4 | **49.4%** | 14520.4 |
| RKV | Base | 28.1% | 19625.3 | 35.1% | 17915.2 | 39.9% | 27429.7 |
| | + Ours | 24.1% | 24346.1 | **52.0%** | 24748.5 | **50.3%** | 21296.5 |

see Section 3.3.1). Due to our method's head- and layer-adaptive nature, actual cache usage remains below Top-$k$ baselines. We default $L_{lower} = L_{upper}/2$.

2. **Dynamic (Top-$p$) Budget.** (Figure 8, Lower) Baselines evict cache only when the token selection probability is zero under the Top-$p$ budget. While this preserves baseline performance, our method regularizes the budget to constrain cache growth, balancing performance with high sparsity for efficiency.

**Baseline Adaptations.** HARD-KV unifies Top-$k$ methods into Top-$p$ head-adaptive counterparts via the 3-step framework (Appendix E). We adapt baselines strictly as logits-level transformations rather than direct algorithmic ports. Compression triggers every 128 steps during decoding, differing from prefill-only settings.

- **VANILLA**: Uses raw attention logits (akin to H2O). For fairness, we enforce the retention of attention sinks and a local window, combining H2O and StreamLLM strategies.

- **SNAPKV**: Max-pools raw logits along the sequence dimension. We adapt the original score-pooling to logit-pooling without affecting selection (Solution 1), while retaining the local window.

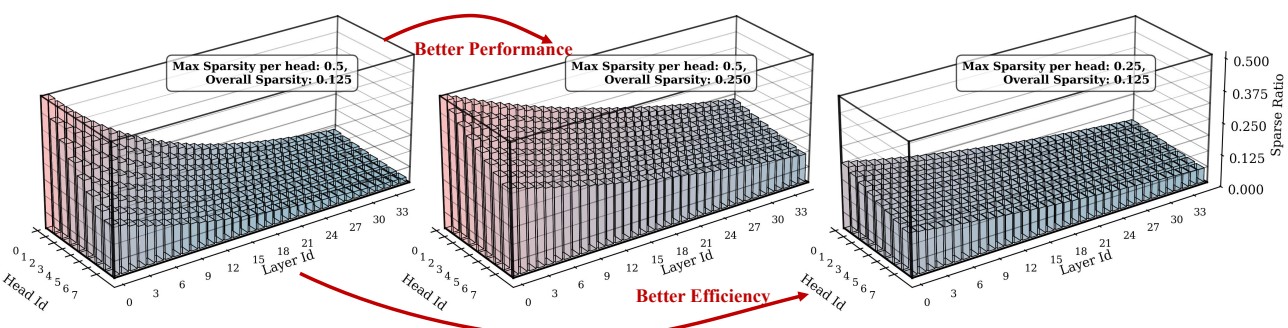

*Figure 9.* **Sparsity Utilization Visualization.** We sort the effective budget for different heads and different layers to draw the distribution. For a lower sparsity utilization (**Left**), the efficiency suffer from the bottleneck in loading unused cache. To improve performance, we can increase sparsity utilization by increasing overall sparsity (regulated by Top-$p$ budget), with approximately same efficiency. To improve efficiency, we can lower the max sparsity per head (regulated by Top-$k$ budget), with same overall sparsity ratio that preserves the performance.

- R-KV: Converts pair-wise similarity computations into a logits-level transformation (Solution 2), similarly preserving attention sinks and the local window.

**Datasets & Models.** We focus on long-decoding scenarios where models generate >10k tokens (e.g., reasoning tasks). We evaluate on AIME24 (Zhang & Math-AI, 2024), AIME25 (Zhang & Math-AI, 2025), and U-Math (Chernyshev et al., 2024) (see Appendix C.1 for details). Experiments use Qwen3-8B (with *think* mode enabled), with Qwen3-4B and Qwen3-32B results provided in Appendix C.3.

### 4.1. Fixed-budget Performance

*Table 2.* **Fixed Budget Performance (Top-$p$).** Comparison of baselines vs HARD-KV across different Top-$p$ budgets ({P70, P80, P90}). The Top-$k$ budget is 4096.

| Method | Base | | Top-$p$ Budget | | | | | |
|---|---|---|---|---|---|---|---|---|
| | Acc | Len | p70 | | p80 | | p90 | |
| | | | Acc | Len | Acc | Len | Acc | Len |
| *Dataset: AIME24* | | | | | | | | |
| Vanilla | 29.7% | 31843.5 | 40.1% | 26495.3 | 23.3% | 27571.8 | 33.5% | 23449.8 |
| SnapKV | 57.7% | 19157.4 | 63.3% | 19386.8 | 56.7% | 21757.3 | 56.7% | 21616.5 |
| RKV | 43.3% | 23281.5 | 53.3% | 27802.4 | 51.5% | 29152.1 | 57.9% | 28094.0 |
| *Dataset: AIME25* | | | | | | | | |
| Vanilla | 24.8% | 31851.4 | 20.1% | 30100.8 | 18.9% | 29155.4 | 20.7% | 28794.4 |
| SnapKV | 27.9% | 25621.4 | 36.6% | 24264.5 | 53.3% | 24229.6 | 42.3% | 25228.6 |
| RKV | 40.0% | 23944.3 | 39.9% | 29174.3 | 46.7% | 28309.9 | 56.7% | 28231.6 |
| *Dataset: UMath* | | | | | | | | |
| Vanilla | 32.1% | 26314.3 | 38.2% | 22502.7 | 43.9% | 17195.4 | 36.1% | 16118.8 |
| SnapKV | 46.7% | 12420.2 | 45.3% | 14212.9 | 48.1% | 14059.4 | 49.4% | 14520.4 |
| RKV | 39.9% | 27429.7 | 38.0% | 23649.9 | 47.5% | 23681.0 | 50.3% | 21296.5 |

**Fixed Budget Results (Top-$k$).** Table 1 presents results under varying Top-$k$ budgets. Our method achieves up to a **1.5×** **performance boost** at the largest budget setting (4096). We observe consistent improvements for both SNAPKV and R-KV across all three datasets. For VANILLA,

while there is a considerable boost in low-budget settings, performance degrades at higher budgets. We attribute this drop to factors intrinsic to the method, specifically the accumulation of noisy signals in raw logits.

A notable finding is the effectiveness of HARD-KV in low-budget regimes, driven by the superior utilization inherent to Top-$p$ sampling. Given the intrinsic difficulty of competition-level math reasoning, Top-$k$ baselines frequently fail in low-budget settings, often yielding overlong, high-perplexity responses (Chen et al., 2025a). By compressing block allocation before the hard budget is reached (Figure 8), our method preserves capacity for retaining critical KV cache states later in the generation process, thereby improving accuracy.

**Fixed Budget Results (Top-$p$).** As discussed above, the fixed Top-$p$ budget acts as a reference for the target token density that compression methods should maintain in the current context, and therefore affects the selections accumulated over compression steps. Table 2 reports performance under different Top-$p$ budgets. For SNAPKV and R-KV, the shifted logits benefit substantially from larger $p$ values, indicating that these adapted methods can dynamically recover useful information from long-tailed attention distributions. By contrast, raw logits (VANILLA) show relatively flat performance across $p$ values, consistent with our earlier observation that VANILLA degrades under larger Top-$k$ budgets. This suggests that simply increasing the retained probability mass is insufficient when the underlying logits contain accumulated noise; addressing this limitation in raw attention logits under high-budget settings is left to future work.

### 4.2. Dynamic-budget Performance

**Dynamic Budget Results.** Table 3 shows performance in dynamic budget settings, along with throughput ablations.

*Table 3.* **Dynamic Budget Trade-offs.** Analysis of Efficiency vs. Accuracy. **SU**: Sparsity Utilization, **Thr.**: Throughput (tokens/s).

| Method | Config | AIME24 | | | AIME25 | | | UMath | | |
|---|---|---|---|---|---|---|---|---|---|---|
| | | Acc | SU | Thr. | Acc | SU | Thr. | Acc | SU | Thr. |
| *System Throughput Baselines (Tokens/s)* | | | | | | | | | | |
| Base System | | | 346 | | | 350 | | | 580 | |
| + CUDA Graph | | | 477 | | | 467 | | | 748 | |
| + Top-$k$ pruning | | | 806 | | | 823 | | | 996 | |
| **Top-$p = 0.90$** | | | | | | | | | | |
| VANILLA | Base | 35.3% | 33.2% | 279 | 33.3% | 35.5% | 289 | 44.5% | 46.6% | 329 |
| | + Ours | 33.5% | 83.4% | 732 | 20.7% | 86.3% | 792 | 36.1% | 79.5% | 819 |
| SNAPKV | Base | 43.3% | 30.7% | 301 | 43.3% | 18.9% | 343 | 42.0% | 37.1% | 321 |
| | + Ours | 49.2% | 90.3% | 632 | 42.6% | 97.9% | 626 | 46.5% | 84.3% | 726 |
| **Top-$p = 0.60$** | | | | | | | | | | |
| VANILLA | Base | 33.9% | 27.1% | 280 | 20.0% | 47.5% | 291 | 25.3% | 40.1% | 317 |
| | + Ours | 36.7% | 77.1% | 728 | 21.0% | 83.1% | 698 | 40.1% | 82.0% | 887 |
| SNAPKV | Base | 26.7% | 23.4% | 300 | 16.7% | 30.1% | 305 | 30.0% | 21.1% | 319 |
| | + Ours | 50.2% | 88.2% | 646 | 43.3% | 85.2% | 645 | 46.0% | 89.2% | 729 |

Under the Top-$k$ setting, both CUDA Graph integrations and pruning contribute crucially to the overall throughput improvements. Under the Top-$p$ setting, baselines here retain tokens whenever any layer or head selects them within the sliding window (see Figure 8 for illustrations). While this yields accurate Top-$p$ selections, it has caused downgraded throughput compared with Top-$k$ counterparts, due to incompatibility with CUDA Graph and blocking operation of compression. Another consequence in naive Top-$p$ pruning is suboptimal sparsity utilization (defined as max sparsity per head divided by overall sparsity; see Figure 9). By the KV index management system introduced in Section 3.3, we can regulate the overall sparsity via a fixed Top-$p$ budget and enforce efficiency via a Top-$k$ threshold. Our method achieves performance comparable to Top-$p$ sparse baselines while maintaining the efficiency of Top-$k$ eviction.

**Summary and Recommendations.** Synthesizing these results, we recommend optimizing performance by regulating sparsity utilization along two dimensions. For performance-sensitive tasks, we suggest using a larger $k$ to increase overall token retention, together with a larger $p$ to improve utilization of the retained cache. For efficiency-sensitive tasks, we suggest using a smaller $k$ to improve throughput, together with a smaller $p$ to reduce occupied cache space and avoid premature eviction.

## 5. Conclusion and Future Work

In summary, we propose a scheme that unifies diverse KV compression methods to facilitate Top-$p$ sampling, incorporating *logits calibration*. Furthermore, we provide a systems-level solution for managing head-adaptive KV indices. By integrating these components, HARD-KV, built upon *Cascade Cache*, achieves a superior trade-off between performance and efficiency compared to fixed-budget baselines.

In future work, we aim to extend the HARD-KV framework to streaming compression regimes (Chen et al., 2025b; Wang et al., 2025b) and investigate sequence-level hybrids of constant-sized and linear-sized memory (Wang et al., 2025a).

## Acknowledgment

This work was supported by the Fundamental and Interdisciplinary Disciplines Breakthrough Plan of the Ministry of Education of China (No. JYB2025XDXM103) and CCF-Ant Research Fund (No. RF20250402).

## Impact Statement

This paper presents work whose goal is to advance the field of Machine Learning. There are many potential societal consequences of our work, none which we feel must be specifically highlighted here.

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

## A. System Design for HARD-INFER

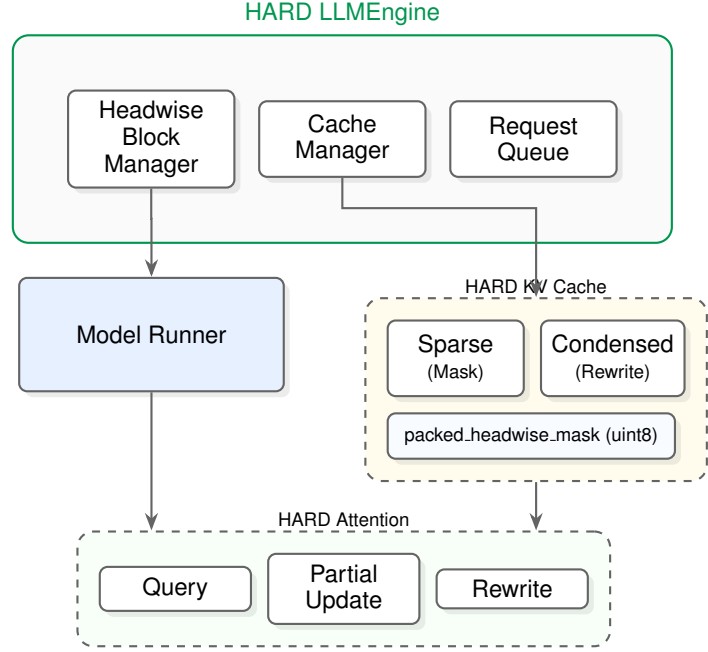

*Figure 10.* nanovllm: Monolithic Dense Cache Architecture.

*Figure 11.* nanovllm_HARD: Hierarchical Sparse-Dense Architecture.

Build upon NANO-VLLM , the HARD-INFER fork have made the following key adaptations:

1. `Block Manager` ⟶ `Headwise Block Manager`
   We implement head-flattened block indices (as shown in Figure 6 in Section 3.3.2). This design facilitates head-adaptive KV block allocation and release, enabling more fine-grained memory management for the KV cache.

2. `Sequence Manager` ⟶ `Cache Manager`
   The vanilla NANO-VLLM manages metadata (such as the `block table`, i.e., allocated KV blocks for a given request) directly via the sequence/request data structure. In contrast, for the **Cascade Cache** in this work, we implement a dedicated `Cache Manager`. We realize the three-tier cache using a unified head-wise mask structure that covers the entire sequence. This head-wise mask is packed in uint8 format for kernel efficiency. The three-tier cache spaces are isolated by pointers in the `req_to_token_pool` (in SGLang forward_batch) and dynamically apply the effective

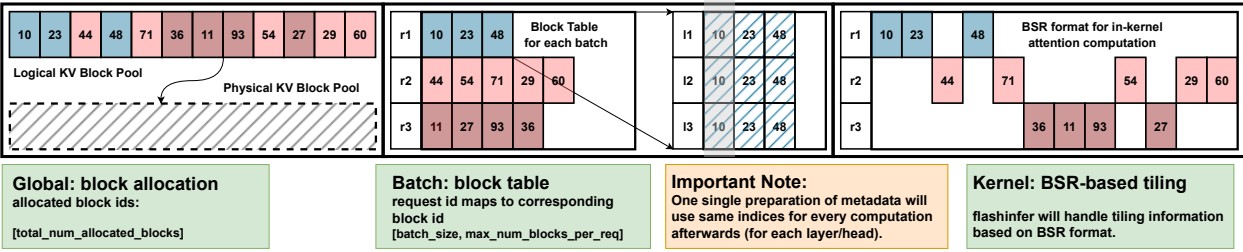

*Figure 12.* The metadata preparations during a complete decoding. **Global:** The logical KV block pointers to the physical KV blocks; **Batch:** The request-to-logical-block mapping, gatherer in a running batch; **Kernel:** The tiling data needed to execute kernel computation, often most computation-intensive.

mask during decoding or KV selection computations.

3. Flash Attention Layer ⟶ HARD Attention Layer

For the attention computation backend, we transition from Flash Attention kernels to FlashInfer. This is due to FlashInfer's native support for a block size of 1, which enables the arbitrary selection of KV indices. To ensure compatibility with CUDA Graphs, we implement a *Partial Update* mechanism that bypasses high-overhead preparation steps to directly update the head-wise mask for different layers. Furthermore, for decoding-time KV cache compression, we store the query cache in a sliding window and support efficient rewrite kernels.

For further details in the architecture of HARD-INFER, please refer to architecture.md in our code repository.

## B. Further Discussions on KV Index Regularization

### B.1. PagedAttention, Page Table, and KV Index Metadata

PagedAttention (Kwon et al., 2023a) addresses memory fragmentation in LLM inference systems. Engines like vLLM (Kwon et al., 2023b) and SGLang (Zheng et al., 2024) implement this via various backends, each necessitating distinct metadata computations.

Decoding workloads are typically memory-bound but suffer disproportionate scheduling overhead. This bottleneck is exacerbated by KV cache compression, particularly under aggressive sparsity ratios (e.g., $\approx 20\%$). The primary source of this overhead is metadata preparation, which we categorize into three levels:

**Global-Level:** The engine pre-allocates a global KV memory pool. At runtime, requests are assigned random block indices (Continuous Batching). While standard attention uses a single index per request, head-adaptive scenarios require dynamic indices across layers. To maintain compatibility with CUDA Graphs, we expand these allocated indices by the number of heads.

**Batch-Level:** Inference executes in batches. For decoding-only requests, we maintain a list of occupied block indices stacked into a block table, often flattened for efficiency. In compression scenarios, this table must support read/write operations to facilitate data selection methods.

**Kernel-Level:** The most computationally intensive phase is tiling metadata for kernel execution. Flashinfer employs a unified Block Compressed Sparse Row (BSR) format to support custom attention mechanisms, including KV compression. To optimize preparation, we employ unified indices across layers; this avoids per-layer overhead while remaining compatible with head-adaptive selection via head expansion. Additionally, we utilize Flashinfer's custom masking to exclude attention scores at specific positions.

### B.2. Discussion on trade-off in the Index Management for Layerwise Selection

In this subsection, we extend the discussion from Section 3.3.1 to explore potential schemas for different KV selection layouts. Head-wise Allocation (HA) requires the most dynamic selection mechanism, while the subsequent three layouts present simpler tasks that can be addressed using similar methodologies. The metrics reported below were collected on an NVIDIA RTX 4090 GPU using a batch size of 32, with an overall sparsity ratio of 12.5% and a maximum per-head sparsity ratio of 25%.

It is important to note that during attention computation, head parallelism can be effectively mapped to query parallelism. Specifically, kernels such as FlashInfer flatten the head dimension, arranging query tile sizes for each Cooperative Thread Array (CTA) via the transformation: $[num\_queries, num\_kv\_heads, head\_dim] \rightarrow [num\_queries \times num\_kv\_heads, 1, head\_dim]$.

**Base Task: Token-wise Selection.** All indices across different layers are identical. We only need to maintain global (batch-level) block indices, requiring no additional optimization efforts.

**Task 1: Layer-wise Selection (LS).** In this setting, while different layers share the same token budget, the specific indices are selected independently.

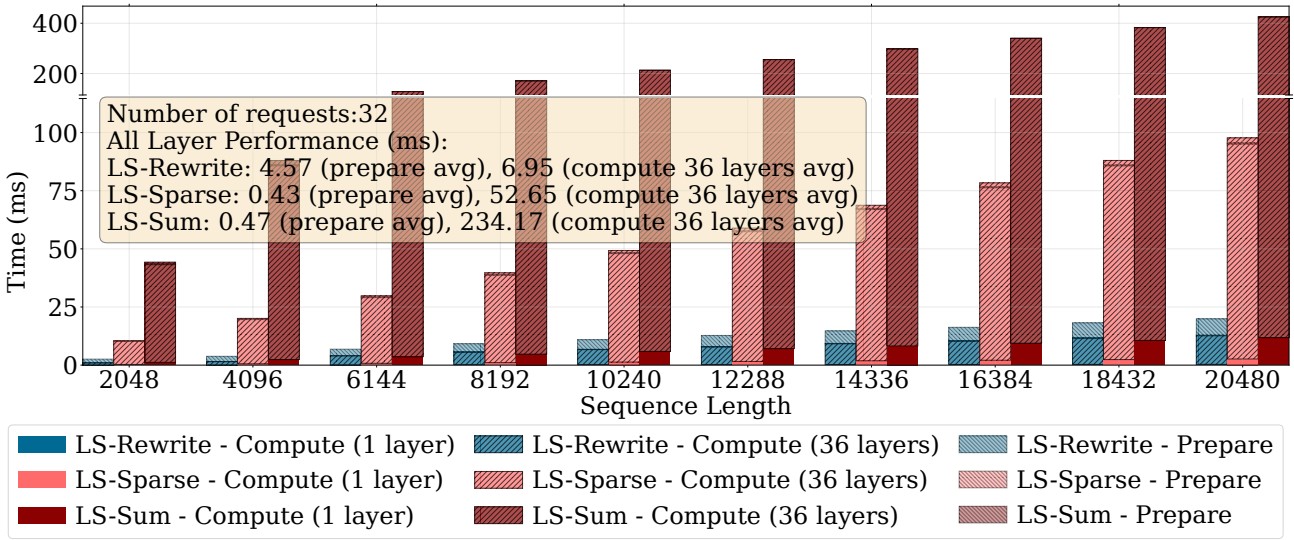

*Figure 13.* Performance evaluations for solutions in the Layer-Selection (LS) task.

**Solutions: Vanilla (✗)** – Requires planning before every layer, resulting in broken CUDA Graphs; **LS-Rewrite (✓)** – Consolidates selected KV blocks by reading and writing them into new blocks with unified indices; **LS-Sparse (✓)** – Maintains full-attention allocated indices and passes a selection mask per layer; **LS-Sum (✗)** – Allocates KV blocks for each layer and concatenates them during attention preparation. This is inefficient due to excessive redundant loading.

**Task 2: Layer-wise Allocation (LA).** Different layers are assigned different budgets with independently selected indices.

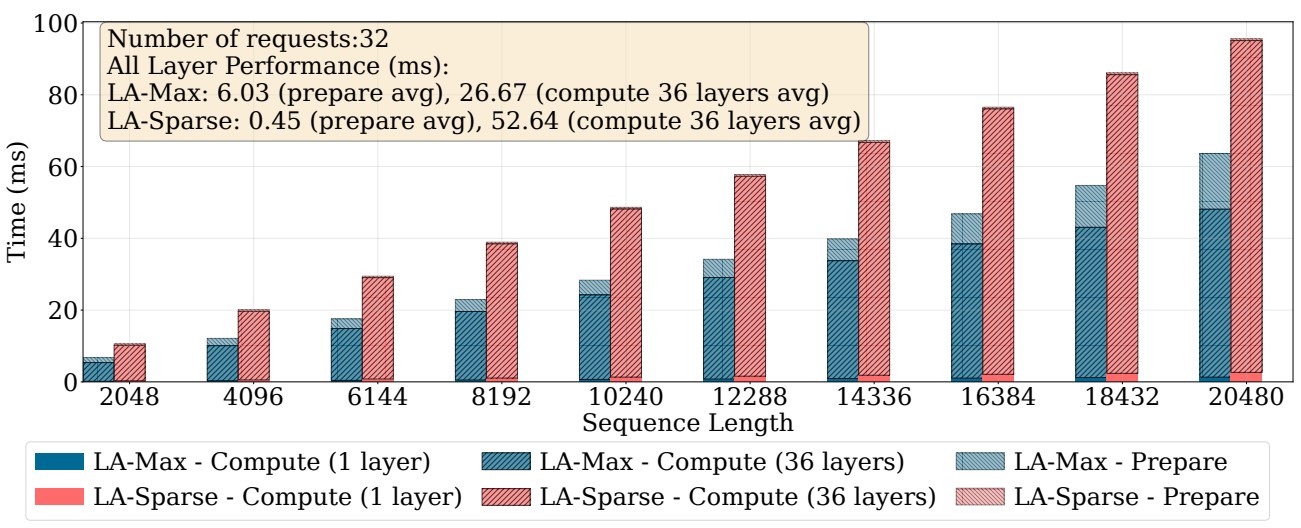

*Figure 14.* Performance evaluations for solutions in the Layer-Allocation (LA) task.

**Solutions: LA-Sparse (–)** – Utilizes sparse loading, identical to 'LS-Sparse'; **LA-Max (✓)** – Unifies block indices based on the maximum layer-wise allocated pages. This necessitates an additional rewrite operation, resulting in higher preparation latency.

*Latency Analysis:* It is important to note that the higher latency incurred by rewriting the KV cache is amortized over decoding steps. We only need to rewrite the cache once, utilizing vanilla allocation thereafter for non-selection decoding steps. For prefill-time compression methods (Li et al., 2024; Xu et al., 2024; Chen et al., 2024a), this additional cost is effectively negligible. Regarding decoding-time compression methods (which use periodic compression operations), the cost is averaged over the compression frequency.

**Task 3: Head-wise Selection (HS).** Different heads have different budgets and independently selected indices, yet they total the same KV cache budget per layer.

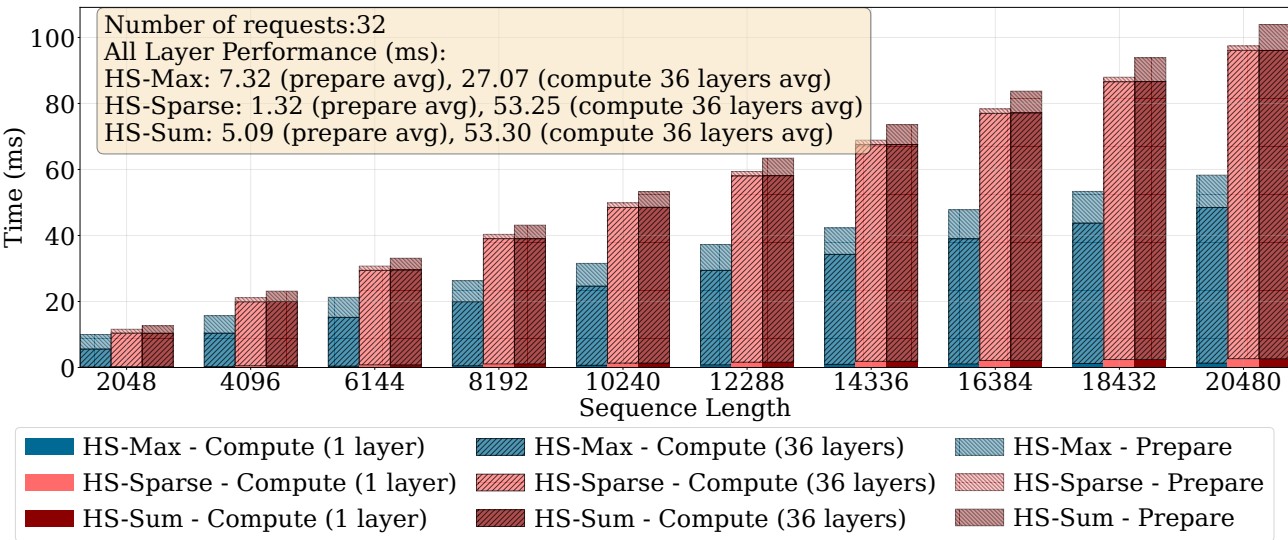

*Figure 15.* Performance evaluations for solutions in Head-Selection (HS) scenarios.

To utilize the benefits of flexible KV cache allocation in this setting, we assign different block IDs to different heads in the global/batch-level metadata. However, we ensure that each block index corresponds to the KV cache of all layers (see Figure 6 in Section 3.3.2).

**Solutions: HS-Sparse (–)** – A sparse loading method that keeps full indices in sequence and loads the effective KV cache via sparse masks; **HS-Max (✓)** – Unifies head-wise indices to the maximum number of allocated blocks per head, requiring a rewrite operation to align indices across heads; **HS-Sum (✗)** – Unifies head-wise indices to the sum of allocated blocks per layer, requiring a complex rewrite operation across layers.

Finally, we reach to **Head-wise Allocation (HA)** task that is discussed in Section 3.3.

**Trade-off in Rewrite Operation**   Building on the trade-off discussed in Section 3.3.2, we characterize the system cost using the Memory-Time-Integral (MTI):

$$MTI(\text{Memory-Time-Integral}) = \int_0^T \text{memory(t)} \, dt$$

Figure 16 compares the MTI of **HA-Sparse** and **HA-Max**. Notably, **HA-Max** quickly offsets the temporary overhead incurred by rewrite operations (in $< 5$ steps). This implies that increasing the size of the *Condense Cache* (**HA-Max**) over the *Sparse Cache* (**HA-Sparse**) improves request-level throughput. Nevertheless, we cannot minimize the Sparse Cache entirely, as it is critical for the *Cautious Update* strategy (Section 3.1). Therefore, we select a balanced memory allocation ($L_{lower} = L_{upper}/2$) for the experimental evaluation presented in Section 3.3.1 and 4.

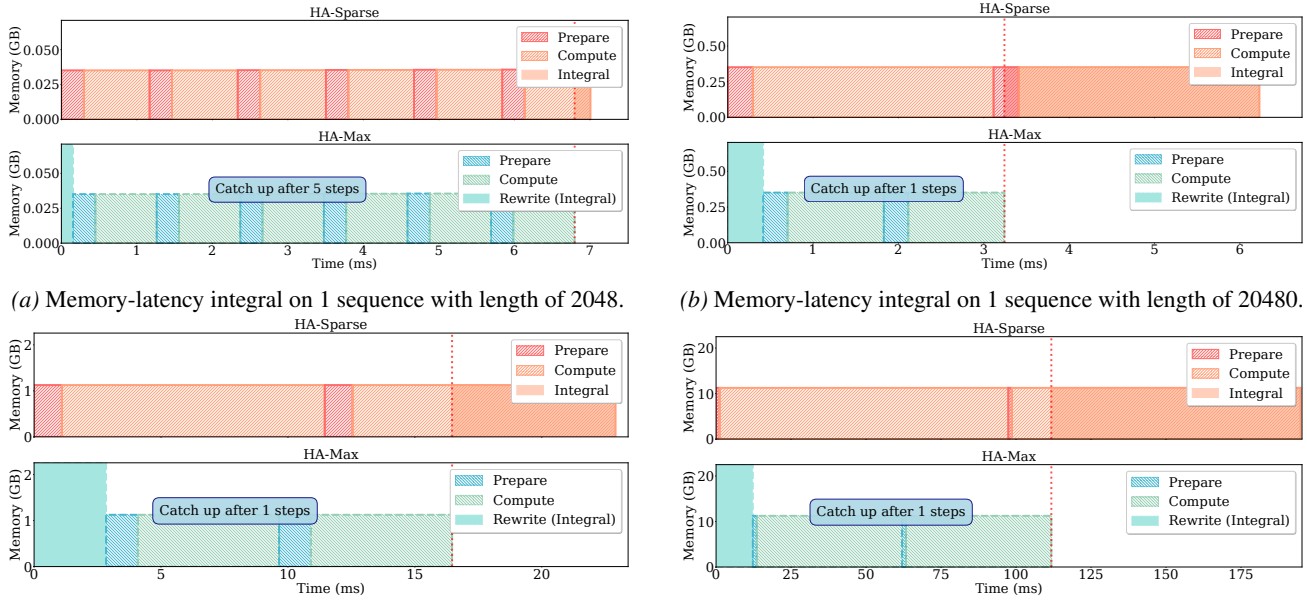

*(a)* Memory-latency integral on 1 sequence with length of 2048.

*(b)* Memory-latency integral on 1 sequence with length of 20480.

*(c)* Memory-latency integral on 32 sequence with length of 2048.

*(d)* Memory-latency integral on 32 sequence with length of 20480.

*Figure 16.* Trade-off measured by Memory-latency integral for different number of sequences and different sequence lengths. In most settings, the rewrite integral can catch up its sparse counterpart in less then 5 decoding steps.

## C. Experiments Details

### C.1. Choice of Datasets

HARD-KV is designed for decoding-time KV cache compression, specifically targeting the challenges inherent in long-context generation. In contrast to traditional long-context tasks—such as document QA, few-shot learning, summarization, and retrieval (Bai et al., 2024; 2025; Hsieh et al., 2024)—which are characterized by *long inputs* and *short outputs*, we focus on scenarios featuring *short inputs* and *long outputs*.

This focus necessitates an evaluation centered on complex mathematical reasoning. Accordingly, we select the **AIME24** (Zhang & Math-AI, 2024), **AIME25** (Zhang & Math-AI, 2025), and **U-MATH** (Chernyshev et al., 2024) benchmarks. U-MATH is a university-level dataset derived from real-world academic sources; it remains relatively under-explored in the literature regarding reasoning models. We find that U-MATH is particularly effective for evaluating KV compression methods, as its moderate overall accuracy allows for a granular distinction between different levels of attention sparsity. For our experimental setup, we filtered the dataset to a subset of 50 non-image-based questions with definitive answers to facilitate objective evaluation. While popular benchmarks such as **MATH-500** (Hendrycks et al., 2021) and **GSM8K** (Cobbe et al., 2021) were considered, they are largely saturated and typically generate fewer than 10k tokens on average. All experimental evaluations were conducted using the `math-verify` (Huggingface, 2025) framework.

### C.2. About Reproduction

All experiments were conducted on NVIDIA A100 and NVIDIA RTX 4090 GPUs. The results reported in Table 1 and Table 2 represent the average of three independent trials for each configuration. For reproducibility, comprehensive guidelines are provided in our documentation. Beyond the code repository, we also release full experiment logs to substantiate the reliability of our results and facilitate further analysis of token generation patterns.

### C.3. Further Experiment Results

We present the experimental results for Qwen3-4B in Table 4 and Table 5. When compared with the Qwen-8B results (Table 1 and Table 2), the performance metrics are approximately equivalent. This similarity can be attributed to the **architectural congruency** between the two models; specifically, Qwen3-4B and Qwen3-8B share **identical** attention layer configurations, possessing the same `head_dim`, `num_heads`, `num_kv_heads`, and `num_layers`. Consequently,

optimizations targeting the KV cache yield consistent effects across both models. This observation aligns with the official technical report Yang et al. (2025), which documents comparable performance profiles for these two architectures.

However, our experiments with Qwen3-4B reveal greater volatility in performance metrics. Notably, we observe instances of improved results associated with longer generation lengths. This finding is counter-intuitive, as extended generation lengths typically degrade performance in the context of KV cache eviction. We leave for future investigation the analysis of how non-attention parameters, such as Feed-Forward Networks (FFN), may modulate the efficacy of KV cache compression methods.

*Table 4.* **Fixed Budget Performance of Qwen3-4B.** Comparison of baselines vs HARD-KV across different Top-$k$ budgets ({1024, 2048, 4096}). The Top-$p$ budget for our methods is set as 0.90. *Full Attention* shows the performance results without budget constraints.

| Dataset | Selection Method | Config | Budget Size | | | | | |
| | | | 1024 | | 2048 | | 4096 | |
| | | | Acc | Gen. Len | Acc | Gen. Len | Acc | Gen. Len |
|---|---|---|---|---|---|---|---|---|
| AIME24 | **Full Attn** | | | | | | 68.2% | 14625.3 |
| | Vanilla | Base | 3.3% | 32768.0 | 36.7% | 28846.5 | 56.7% | 18706.9 |
| | | + Ours | **24.0%** | **31790.8** | **39.9%** | **30273.2** | **36.6%** | **30864.9** |
| | SnapKV | Base | 3.2% | 32652.4 | 36.0% | 29044.4 | 36.7% | 19007.7 |
| | | + Ours | **10.7%** | **32768.0** | **40.2%** | **29201.6** | **55.4%** | **20931.6** |
| | RKV | Base | 3.3% | 26750.1 | 26.7% | 26245.9 | 50.0% | 18967.4 |
| | | + Ours | **23.4%** | **31036.7** | **43.3%** | **31788.0** | **56.5%** | **28267.2** |
| AIME25 | **Full Attn** | | | | | | 63.3% | 17998.7 |
| | Vanilla | Base | 6.5% | 32309.7 | 16.7% | 26617.5 | 24.8% | 31851.4 |
| | | + Ours | **10.5%** | **30906.7** | **12.9%** | **30837.8** | **20.7%** | **29966.5** |
| | SnapKV | Base | 10.9% | 32229.7 | 12.3% | 30040.4 | 43.3% | 23654.5 |
| | | + Ours | **9.3%** | **32768.0** | **26.7%** | **28254.2** | **44.3%** | **26548.9** |
| | RKV | Base | 13.7% | 27924.7 | 21.9% | 26836.2 | 30.7% | 20616.6 |
| | | + Ours | **13.2%** | **31496.4** | **20.7%** | **31809.6** | **53.4%** | **28202.8** |
| UMath | **Full Attn** | | | | | | 53.3% | 10111.5 |
| | Vanilla | Base | 14.0% | 28859.6 | 36.7% | 19021.4 | 44.0% | 12828.0 |
| | | + Ours | **36.7%** | **23695.8** | **36.2%** | **24691.8** | **38.0%** | **24832.7** |
| | SnapKV | Base | 18.0% | 29332.5 | 31.8% | 20651.4 | 44.2% | 14371.1 |
| | | + Ours | **22.5%** | **29204.8** | **42.0%** | **23663.2** | **49.4%** | **16209.9** |
| | RKV | Base | 30.1% | 18504.6 | 35.1% | 17685.8 | 52.0% | 12825.3 |
| | | + Ours | **20.1%** | **31351.6** | **42.0%** | **23663.2** | **50.3%** | **23156.9** |

We further report results for Qwen3-32B in Table 6. Serving larger-scale models poses substantial system challenges: on a single GPU, such models leave less than 20GB for the KV cache (e.g., Qwen3-32B on a single NVIDIA A100 80GB GPU), making multi-GPU parallelism necessary. We adopt Tensor Parallelism (TP), rather than Pipeline Parallelism, which introduces pipeline bubbles, or Expert Parallelism, which is incompatible with CUDA Graphs.

Since TP naturally partitions attention heads across instances, each instance performs head-wise KV eviction independently. To minimize communication overhead, our customized NANO-VLLM engine implements two key optimizations:

- **Asynchronous mask management.** We avoid synchronizing head-wise sparse masks across instances by managing them on the CPU, thereby preventing frequent CPU–GPU synchronization stalls.

- **Localized metadata.** Sequence-level metadata, such as head-wise block tables, is stored locally within each instance, eliminating cross-instance serialization and communication overhead.

As a result, the only required communication is a one-time gather of head counts per sequence, which introduces negligible overhead compared to non-parallel configurations.

*Table 5.* **Dynamic Budget Performance of Qwen3-4B.** Comparison of baselines vs HARD-KV across different Top-$p$ budgets ({P70, P80, P90}). The Top-$k$ budget is 4096.

| Dataset | Method | Base | | Top-$p$ budget | | | | | |
| | | Acc | Gen. Len | p70 | | p80 | | p90 | |
| | | | | Acc | Gen. Len | Acc | Gen. Len | Acc | Gen. Len |
|---|---|---|---|---|---|---|---|---|---|
| **AIME24** | Vanilla | 56.7% | 18709.9 | **28.5%** | **31004.0** | **22.0%** | **31453.0** | **36.6%** | **30864.9** |
| | SnapKV | 36.7% | 19007.7 | **50.0%** | **23631.7** | **56.7%** | **29990.4** | **66.7%** | **21796.8** |
| | RKV | 50.0% | 18967.4 | **53.3%** | **27802.4** | **51.5%** | **29152.1** | **56.5%** | **28267.2** |
| **AIME25** | Vanilla | 24.8% | 31851.4 | **17.2%** | **29977.0** | **26.7%** | **30941.2** | **20.7%** | **29966.5** |
| | SnapKV | 43.3% | 23654.5 | **50.0%** | **22225.1** | **40.0%** | **27711.6** | **44.3%** | **26548.9** |
| | RKV | 30.7% | 20616.6 | **36.9%** | **28183.2** | **40.3%** | **28214.1** | **53.4%** | **28202.8** |
| **UMath** | Vanilla | 44.0% | 12828.0 | **42.0%** | **25278.8** | **39.3%** | **25729.9** | **38.0%** | **24832.7** |
| | SnapKV | 44.2% | 14371.1 | **44.6%** | **15291.4** | **52.3%** | **16546.8** | **49.4%** | **16209.9** |
| | RKV | 52.0% | 12825.3 | **50.0%** | **22459.1** | **52.0%** | **22564.0** | **50.3%** | **23156.9** |

*Table 6.* **Fixed Budget Performance of Qwen3-32B (Budget: 4096).** Transposed comparison of baselines vs HARD-KV for a Top-$k$ budget of 4096 across datasets. The Top-$p$ budget for our methods is set as 0.90. *Full Attention* shows performance results without budget constraints.

| Selection Method | Config | AIME24 | | AIME25 | | UMath | |
| | | Acc | Gen. Len | Acc | Gen. Len | Acc | Gen. Len |
|---|---|---|---|---|---|---|---|
| **Full Attn** | | 76.7% | 13104.0 | 70.0% | 16852.0 | 56.0% | 9169.0 |
| Vanilla | Base | 40.0% | 25918.0 | 33.3% | 28701.0 | 48.0% | 22183.0 |
| | **+ Ours** | **50.0%** | **22164.0** | **43.3%** | **24474.0** | **52.0%** | **14711.0** |
| SnapKV | Base | 53.3% | 24862.0 | 33.3% | 24256.0 | 50.0% | 21097.0 |
| | **+ Ours** | **60.0%** | **21549.0** | **46.7%** | **22901.0** | **56.0%** | **21778.0** |
| RKV | Base | 60.0% | 19911.0 | 40.0% | 21275.0 | 50.0% | 10537.0 |
| | **+ Ours** | **66.7%** | **17030.0** | **50.0%** | **21427.0** | **52.0%** | **10123.0** |

# D. Further Discussions on logits Calibration

### D.1. Algorithms for solving Problem 1 under order-invariance constraint

In this section, we will provide the two choices of algorithms to solve Problem 1 under the order-invariance Constraint 1.

Algorithm 1 uses Gradient Descend to optimize temperatures $T$ as parameters. This algorithm treats the Problem 1 as an optimization problem that can be solved by modern optimizers (Kingma & Ba, 2014; Loshchilov & Hutter, 2017; Ruder, 2016). In practice, we prefer Adam (Kingma & Ba, 2014) which converge quicker and have stable performances across different settings.

Algorithm 2 views Problem 1 as a searching problem, when the transformation function satisfy certain constraints (like being concave), multiplying logits $z$ by $\frac{1}{T}$ yields sharpened distribution for lower $T$ and flattened distribution for higher $T$. The sharpened/flattened distribution can be this way shaped to satisfy the mass-preserving in Problem 1

In practice, we apply Algorithm 1 for both SNAPKV and R-KV. An example of solved temperatures $T$ is shown in Figure 17. The results is largely constant in different decoding steps, while failure cases exist because of numerical instability and the illness of the problem. For this reason, we only solve the temperatures for the first compression in each request, since solving temperatures in every decoding step is not necessary.

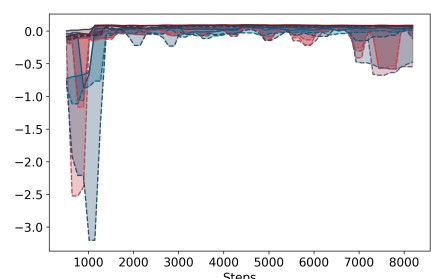

*Figure 17.* An example of solved temperatures for R-KV. The solution of temperatures are relatively stable despite exceptional failures.

---

**Algorithm 1** Gradient Descent on $T$

---

**Parameters :** $k$: for Top-$k$; $p$: for Top-$p$;
$\qquad\qquad$ $T$: temperatures with shape
$\qquad\qquad$ $[bsz \times num\_kv\_heads, query\_len]$
**Optimizers :** $Optimizer$ (e.g. AdamW);
**Initialize** $\quad$ : $T, Optimizer$

1 $k \leftarrow Num\_Selected\_TopP(S, p)$
2 **while** $\|M_k(S) - M_k(\hat{S})\| < \mathcal{E}$ **and** $i \leq N$ **do**

| Solution 1 | Solution 2 |
|---|---|
| 3 $\qquad\hat{z} \leftarrow g(z/T)$ | $\qquad\hat{z} \leftarrow g(z)$ |
| $\qquad\hat{S} = \text{Softmax}(\hat{z})$ | $\qquad\hat{S} = \text{Softmax}(\hat{z}/T)$ |

4 $\quad \ell \leftarrow \|M_s(S) - M_k(\hat{S})\|$ Update $T$ via $Optimizer$

---

**Algorithm 2** Binary Search on $T$

---

**Parameters :** $k$: for Top-$k$; $p$: for Top-$p$;
$\qquad\qquad$ $T$: temperatures $[bsz \times num\_kv\_heads, query\_len]$
**Initialize** $\quad$ : $T_{min}, T_{max}$

1 $k \leftarrow Num\_Selected\_TopP(S, p)$
2 **do**
3 $\quad T_{mid} \leftarrow \frac{T_{min}+T_{max}}{2}$
4

| Solution 1 | Solution 2 |
|---|---|
| $\qquad\hat{z} \leftarrow g(z/T_{mid})$ | $\qquad\hat{z} \leftarrow g(z)$ |
| $\qquad\hat{S} \leftarrow \text{Softmax}(\hat{z})$ | $\qquad\hat{S} \leftarrow \text{Softmax}(\hat{z}/T_{mid})$ |
| $\quad T_{min} \leftarrow \text{WHERE}(M_k(S) < M_k(\hat{S}), T_{mid}, T_{min})$ | $\quad T_{max} \leftarrow \text{WHERE}(M_k(S) < M_k(\hat{S}), T_{max}, T_{mid})$ |

5 **while** $\|M_k(S) - M_k(\hat{S})\| < \mathcal{E}$ **and** $\text{ALL}(T_{min} < T_{max})$;

---

### D.2. Proof of order-preserving temperature transformation

In this part, will prove that when applying function (like max-pool) on the raw logits $z$ do not affect the Top-$k$ selection when applied on the attention score $S$ (order-preserving), we can shape the attention distribution by directly conduct temperature transformation $z/T$ without changing the selection results. Let $z$ denote the attention logits and $S(\cdot) = \text{Softmax}(\cdot)$. We consider a pooling or aggregation function $g(\cdot)$ (such as $\text{MaxPool}$) applied to chunks of the input. We define $\hat{S}$ as the altered attention score derived from the logits.

**Definition D.1** (Monotonic Consistency). Let $g(\cdot)$ be a function such that the ranking of its outputs is preserved under element-wise monotonic transformation. Specifically, $g$ satisfies the following condition:

**Condition 1.** *For any two inputs $z_{(1)}$ and $z_{(2)}$, if $S\big(g(z_{(1)})\big) \geq S\big(g(z_{(2)})\big)$, then:*

$$g(S(z_{(1)})) \geq g(S(z_{(2)}))$$

This condition holds for order-statistic functions such as $\text{MaxPool}$, $\text{MinPool}$, and $\text{Median}$, but not necessarily for arithmetic means.

**Lemma D.2.** *For any temperature $T > 0$, the scaling function $f(x) = x/T$ is strictly monotonically increasing. Consequently, for any $a, b \in \mathbb{R}$:*

$$a \geq b \iff a/T \geq b/T$$

*Proof.* The proof follows from the definition of a strictly increasing function on strictly positive denominators. $\qquad\square$

**Theorem D.3.** *Let $\text{TopK}(\cdot)$ be the operator that returns the set of indices corresponding to the $k$ largest values. If $g(\cdot)$ satisfies Condition 1, then the selection of key indices is invariant to temperature scaling $T > 0$ and the order of Softmax*

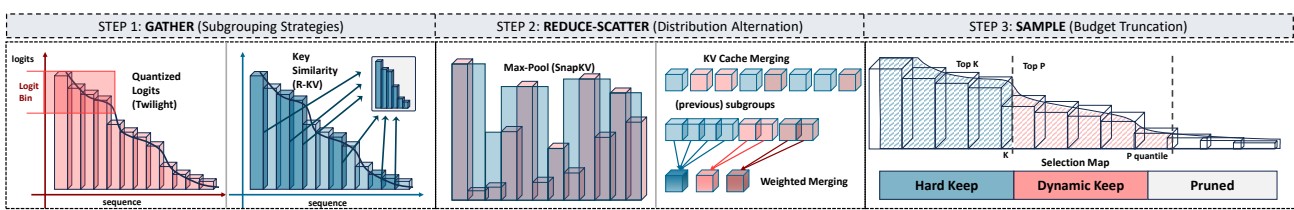

*Figure 18.* (Step 1) **Gather** the KV cache in subgroups along the sequence for following computation; (Step 2) **Reduce** in the subgroup to calculate ranking score and **scatter** to the post-compressed blocks; (Step 3) **Sample** by the scattered score to satisfy Top-$k$ or Top-$p$ constraints.

*application. Formally:*

$$\hat{S}|_k \equiv g(z)|_k$$

*where $\hat{S} = S(g(z/T))$, $|_k$ represent the Top-k subset.*

*Proof.* We aim to show that the ranking of elements in $\hat{S}$ is identical to the ranking of elements in $g(z)$.

Consider two arbitrary indices $(1)$ and $(2)$ such that the pooled logit of $(1)$ is greater than or equal to $(2)$:

$$g(z_{(1)}) \geq g(z_{(2)})$$

**Step 1: Invariance to Temperature (Lemma D.2)**
Since $T > 0$, applying Lemma D.2:

$$g(z_{(1)})/T \geq g(z_{(2)})/T \implies g(z_{(1)}/T) \geq g(z_{(2)}/T)$$

(Note: Since $g$ is order-preserving, $g(cx) = cg(x)$ for $c > 0$ in the case of MaxPool, or generally preserves rank).

**Step 2: Monotonicity of Softmax**
The Softmax function $S(\cdot)$ is strictly monotonic with respect to its input arguments. Therefore:

$$g(z_{(1)}/T) \geq g(z_{(2)}/T) \iff S\big(g(z_{(1)}/T)\big) \geq S\big(g(z_{(2)}/T)\big)$$

Let $\hat{S} = S(g(z/T))$. We have established that the order of $\hat{S}$ is determined strictly by the order of $g(z)$.

**Step 3: Generalization via Condition 1**
Using Condition 1, we ensure that this ranking is consistent with the "true" attention distribution (had Softmax been applied before pooling). Since $S(g(\cdot))$ preserves the order of $g(\cdot)$, and Condition 1 links this to $g(S(\cdot))$, we conclude that identifying the Top-K elements in the computationally efficient proxy $\hat{S}$ yields the same indices as the original computation.

Thus, the Top-K indices remain invariant. □

# E. The Formal Definitions of the Three-step Framework for KV Selection

Let $S_{(i)} = S(\mathbf{q}, \mathbf{k}_{(i)})$ denote the attention score for a query $\mathbf{q}$ and key $\mathbf{k}_{(i)}$. The framework proceeds as follows:

**Gather**: We define a set of indices $G_i$ based on a grouping function $f$ (e.g., quantization or similarity) and a corresponding equivalence relation:

$$G_i = \{j \mid f(k_j, S_i) \sim f(k_i, S_i)\}. \tag{1}$$

**Reduce-Scatter**: We calculate a proxy score $\hat{S}_i$ for the group and scatter it to the associated indices:

$$
\begin{aligned}
\hat{S}_i &= \text{REDUCE}(\{S(q, k_j) \mid j \in G_i\}) \\
\text{SCATTER}&(S_j := \hat{S}_i \mid j \in \hat{G}_i).
\end{aligned}
\tag{2}
$$

Note that the reduction function may operate on the keys $k_i$, necessitating the re-computation of $\hat{S}_i$. In KV cache compression scenarios, the original subgroup $G_i$ need not satisfy $\hat{G}_i \equiv G_i$, where $\hat{G}_i$ represents a compressed subgroup (occupying less memory).

**Sample**: The final set of indices $\mathcal{I}_{\text{final}}$ is determined by:

$$\mathcal{I}_{\text{final}} = \text{SELECT}(\{\hat{S}_i\}), \tag{3}$$

where $\text{SELECT} \in \{\text{Top-}k, \text{Top-}p\}$.

We perform logit calibrations (Section 3.2) after Step 2, **Reduce-Scatter**, which solve the Problem 1 by any one of the above Algorithm 1 and 2. The solution process, can be alternatively conducted in subgroups generated in Step 1, **Gather**, bringing better efficiency in optimizing on temperatures $T$. For the reason stated in Appendix D.1, we reuse the temperatures solved in the first compression step.

## F. Mathematical Abstraction of Cascade Attention

To seamlessly compute attention across these physically separated tiers, we can utilize a composable Attention State algebra proposed by Flashinfer (Ye et al., 2025) originally designed for Prefix Caching (Ye et al., 2024).

Recall that standard self-attention for a query $\mathbf{q}$ and a set of KV pairs indexed by $I$ is computed as:

$$\text{Attention}(\mathbf{q}, I) = \frac{\sum_{i \in I} \exp(\mathbf{q}\mathbf{k}_{(i)}^{\top})\mathbf{v}_{(i)}}{\sum_{j \in I} \exp(\mathbf{q}\mathbf{k}_{(j)}^{\top})}. \tag{4}$$

The denominator represents the total attention mass, which we term the *Sum of Exponentials* ($\text{SE}(I)$). The numerator represents the unnormalized weighted sum of value vectors. $\mathbf{o}(I)$ is defined as the attention result restricted solely to the subset $I$:

$$\mathbf{o}(I) = \sum_{i \in I} \text{Softmax}(\text{SE}_{(i)})\mathbf{v}_{(i)} = \frac{\sum_{i \in I} \text{SE}_{(i)} \, \mathbf{v}_{(i)}}{\text{SE}(I)}, \tag{5}$$

We define the **Attention State** of a tier $I$ as the tuple $\begin{bmatrix} \mathbf{o}(I) \\ \text{SE}(I) \end{bmatrix}$, which fully encapsulates the partial computation of $I$. To aggregate results from multiple tiers (e.g., a Sparse Tier $I$ and a Dense Tier $J$), a binary **Merge Operator** $\oplus$ can be applied to merge the partial outputs for the complete outputs:

$$\begin{bmatrix} \mathbf{o}(I \cup J) \\ \text{SE}(I \cup J) \end{bmatrix} = \begin{bmatrix} \mathbf{o}(I) \\ \text{SE}(I) \end{bmatrix} \oplus \begin{bmatrix} \mathbf{o}(J) \\ \text{SE}(J) \end{bmatrix}$$
$$= \begin{bmatrix} \frac{\mathbf{o}(I)\,\text{SE}(I) + \mathbf{o}(J)\,\text{SE}(J)}{\text{SE}(I) + \text{SE}(J)} \\ \text{SE}(I) + \text{SE}(J) \end{bmatrix}. \tag{6}$$

## G. A Case Study on Attention Property-Aware KV Compression on the Calibrated Logits

The *Logits Calibration* proves to be capable of preserving the density of selection heatmap. In this section, we will provide another example that preserves Attention property (like density), built on top of *Logits Calibration*.

**LSE-preserving KV Cache Merging** Recall that LSE is defined as:

$$\text{LSE}(q, K) = \log\left(\sum_j e^{qK_{(j)}^{\top}}\right)$$

This metric is computed by reduction over the whole sequence and is yielded during the computation of Flash Attention. As shown in Figure 19, LSE connects closely with the

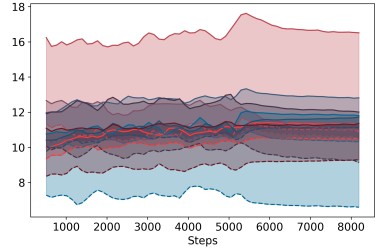

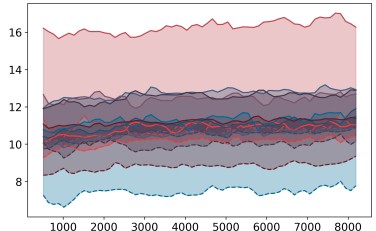

*(a)* The LSE of the compressed KV under budget of 512 (SnapKV).

*(b)* The LSE of the compressed KV under budget of 512 (R-KV).

*Figure 20.* The two figured above show the drop in LSE and a growing magnitude as the seuqence grows.

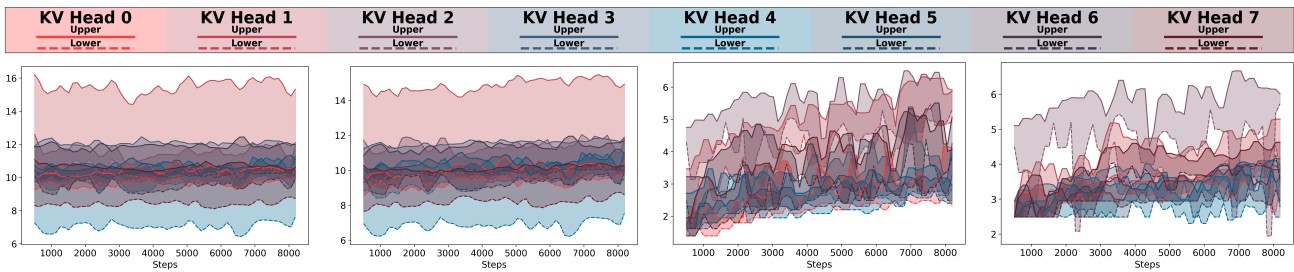

*(a)* LSE for raw attention scores.    *(b)* P60 LSE for raw attention scores.    *(c)* P60 log number of selected tokens (raw).    *(d)* P60 log number of selected tokens (maxpool-ed).

*Figure 19.* LSE and log number of selected tokens. **Observation 1:** (comparing (a) and (b)), a drop in log scale when calculating LSE for a given $p$ compared with full counterparts; **Observation 2:**(comparing (a) and (c)), an increasing trend for both LSE and log number of selections. **Observation 3:**(comparing (c) and (d)), a clear drop in log scale in selections by maxpool-ed logits.

number of selected tokens (density) in log scale.

This intrigues us to investigate if the LSE for the kept tokens can still keep the same trend, when executing strict Top-$k$ compression (eviction for both methods), If the trend remains stable, this suggest the possibility of constant memory; otherwise the property of oracle attention is impaired, especially in the our specific scenario, where KV cache are periodically compressed to fixed budget during long-context decoding.

As shown in Figure 20, both methods have encountered clear drop in LSE which accumulates continually as the sequence grows. Intuitively, LSE serves as a health indicator for attention computation (as it is soft argmax operator) and should reflect or assess the illness of a periodic compression methods. For both TopK selection, we have observed loss in LSE (in Figure 20) , which leads to discussion in the next section on how to preserve LSE when pruning KV Cache.

Denote $\mathrm{SE}(q, K) = \sum_j e^{qK_{(j)}^\top}$, we have:

$$O_{oracle} = \sum_i \frac{e^{qK_{(i)}^\top} V_{(i)}}{\mathrm{SE}(q, K)}$$

$$O_1 = \left( O_{oracle} - \frac{e^{qK_{(j)}^\top} V_{(j)} + e^{qK_{(k)}^\top} V_{(k)}}{\mathrm{SE}(q, K)} \right) \times \frac{\mathrm{SE}(q, K)}{\mathrm{SE}\left(q, K \backslash \left\{K_{(j)}, K_{(k)}\right\} \cup \left\{f(K_{(j)}, K_{(k)})\right\}\right)} \quad (7)$$

$$+ \frac{e^{qf(K_{(j)}^\top, K_{(k)}^\top)} g(V_{(j)}, V_{(k)})}{\mathrm{SE}\left(q, K \backslash \left\{K_{(j)}, K_{(k)}\right\} \cup \left\{f(K_{(j)}, K_{(k)})\right\}\right)}$$

$O_{oracle}$ is the precise output of the self-attention. For the initial compression, we denote the compressed output as $O_1$ (will be extended to $O_i$ in further compressions), the concept of the "first" compression is limited to the selection of two pairs of KV cache for demonstration. This single step of compression does not have to make a complete operation. It is only the **minimal unit** that we can analyze the intervention on the attention output for **selective compression** methods. The $K_{(j)}$ $(V_{(j)})$ represent the $j$-th key (value) vectors in the whole sequence (the index in sequence will be wrapped in bracelets by default). For each specific compression method, we formalize by pair-wise mapping, i.e. $f(K_{(j)}, K_{(k)}) := (K_{(j)}, K_{(k)}) \to \hat{K}_{(j)}$. For evictions, $f$ is formularized as $f(K_{anchor}, K_{evict}) = \mathbb{0} \cdot K_{anchor} + \mathbb{1} \cdot K_{evict}$; for merging methods, $f$ can be $f(K_{(j)}, K_{(k)}) = w_1 \cdot K_{(j)} + w_2 \cdot K_{(k)}$ or in a more complex form.

In the following deduction, we will simplify some representations for ease of notations. $K$ and $V$ will be the current key/value vectors in the $i$-th compression steps, and $\hat{K}$ and $\hat{V}$ will be in the $(i + 1)$-th step, i.e. $\hat{K} = f(K_{(j)}, K_{(k)})$ $(\hat{V} = g(V_{(j)}, V_{(k)}))$. $O_i$ and $\mathrm{SE}_i$ represent the $i$-th reduction result (without bracelets).

$$O_i = \left( O_{i-1} - \frac{e^{qK_{(j)}^\top} V_{(j)} + e^{qK_{(k)}^\top} V_{(k)}}{\mathrm{SE}_{i-1}} \right)$$

$$\times \frac{\mathrm{SE}_{i-1}}{\mathrm{SE}_i} + \frac{e^{q\hat{K}^\top} \hat{V}}{\mathrm{SE}_i}$$

$$\Delta O_i = O_i - O_{i-1}$$

$$= \underbrace{\left( \frac{e^{qK_{un}^\top} V_{un}}{\mathrm{SE}_i} - \frac{e^{qK_{un}^\top} V_{un}}{\mathrm{SE}_{i-1}} \right)}_{\text{Shift in Unchanged Tokens}} \tag{8}$$

$$+ \underbrace{\left( \frac{e^{q\hat{K}^\top} \hat{V}}{\mathrm{SE}_i} - \frac{e^{qK^\top} V}{\mathrm{SE}_{i-1}} \right)}_{\text{Shift due to Compression}}$$

**Theorem G.1.** *Denote a proper linear approximation (weighted attention score) of key cache merging as the following :*

$$\hat{K}_{(new)} = \frac{e^{qK_{(j)}^\top}}{e^{qK_{(j)}^\top} + e^{qK_{(k)}^\top}} K_{(j)} + \frac{e^{qK_{(k)}^\top}}{e^{qK_{(j)}^\top} + e^{qK_{(k)}^\top}} K_{(k)} \tag{9}$$

*The solution to minimize the following:*

$$\Delta \mathrm{SE}_i = \left| e^{q\hat{K}_{(new)}^\top} - \left( e^{qK_{(j)}^\top} + e^{qK_{(k)}^\top} \right) \right| \quad (\text{SE } preserving) \tag{10}$$

*is to let*

$$qK_{(j)}^\top \gg qK_{(k)}^\top. \tag{11}$$

*Proof.* Suppose $\Delta \mathrm{SE}_i \equiv 0$, we have the following system of equations:

$$\begin{cases} q\hat{K}^\top = q \left( \dfrac{e^{qK_{(j)}^\top} K_{(j)} + e^{qK_{(k)}^\top} K_{(k)}}{e^{qK_{(j)}^\top} + e^{qK_{(k)}^\top}} \right)^\top & (9.a) \\[3ex] e^{q\hat{K}^\top} = e^{qK_{(j)}^\top} + e^{qK_{(k)}^\top} & (9.b) \end{cases}$$

Let $qK_{(j)}^\top = x_1$ and $qK_{(k)}^\top = x_2$, combining Equation 9.a and 9.b, we get the following:

$$\frac{x_1 \ln x_1 + x_2 \ln x_2}{x_1 + x_2} = \ln (x_1 + x_2) \tag{13}$$

Rearranging this leads to the *Entropy* of the attention distribution (which is always non-negative).

$$\underbrace{x_1 \ln \left( 1 + \frac{x_2}{x_1} \right) + x_2 \ln \left( 1 + \frac{x_1}{x_2} \right)}_{\text{Always} > 0} = 0. \tag{14}$$

Stated algebraically, let $t = \frac{x_2}{x_1}$, we have $h(t) = \ln(1 + t) + t \ln(1 + \frac{1}{t}) > 0$ (Jensen Inequality). We would like to solve the condition for $t$ that leads to $h(t) \to 0$. We have:

$$\lim_{t \to 0} h(t) = 0$$

$$\text{with } t = \frac{x_2}{x_1} = \frac{e^{qk_2^\top}}{e^{qk_1^\top}} = e^{q(k_2 - k_1)^\top} \tag{15}$$

$$\Rightarrow q(k_2 - k_1)^\top \to -\infty$$

$$\implies qk_1^\top \gg qk_2^\top$$

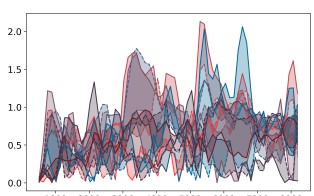 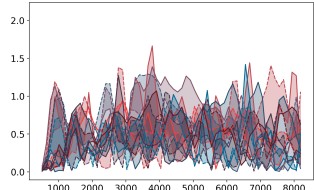 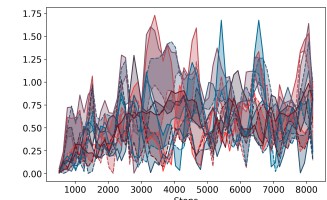 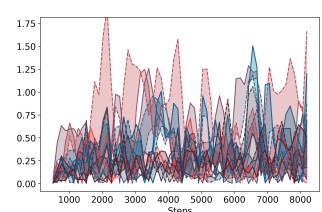

*(a)* The LSE absolute error of SNAPKV pruned KV cache (fixed 512 budget) v.s. full KV cache.

*(b)* The LSE absolute error of SNAPKV pruned-and-merged KV cache (fixed 512 budget) v.s. full KV cache.

*(c)* The LSE absolute error of R-KV pruned KV cache (fixed 512 budget) v.s. full KV cache.

*(d)* The LSE absolute error of R-KV pruned-and-merged KV cache (fixed 512 budget) v.s. full KV cache.

*Figure 21.* The absolute error comparison between w. and w.o. LSE-preserved merging for SNAPKV and R-KV

As shown in the figure 21a 21b 21c and 21d, with LSE-preserved merging, the tendency of increasing absolute error is clearly suppressed. We will provide a more thorough analysis in the Appendix to explain why pruning long-tailed KV cache will result in a growing absolute error and why LSE-preserve merging can suppress such tendency.

