# OpenReview forum: "HARD-KV: Head-Adaptive Regularization for Decoding-time KV Compression"
_ICML.cc/2026/Conference — ICML 2026 regular_

### Official Review · Reviewer_5uEu · 2026-03-09

**Soundness:** 3
**Presentation:** 3
**Significance:** 3
**Originality:** 3
**Overall Recommendation:** 4
**Confidence:** 3

**Summary:**

This paper proposes HARD-KV, a framework to address the challenge that the size of KV grows tremendously during decoding of modern decoder-only LLMs. To alleviate the KV cache burden of LLM deployment, the authors introduces a 3 level cascade cache (dense, sparse, and condensed) to enable a larger decoding batch size by limiting the on-GPU KV cache block number. Furthermore, this paper theoretically unifies the top-k and top-p sparse attention (KV selection) by Logits Calibration. System-level optimizations, such as cache rewriting, sparse loading, and index flattening are introduced to fully unlock the efficiency of the proposed algorithm. Sufficient empirical results show that HARK-KV can outperform existing SOTA methods, e.g. SnapKV and RKV, in terms of task performance, while enhancing the decoding efficiency significantly.

**Compliance With Llm Reviewing Policy:**

Affirmed.

**Final Justification:**

All weaknesses and questions resolved. I'm keeping my positive score.

**Key Questions For Authors:**

See "weaknesses".

**Limitations:**

yes

**Strengths And Weaknesses:**

## Strength

---

1. The motivation of this paper is well-stated and meaningful. The proposed method is effective in terms of both task performance and decoding efficiency.

2. The proposed algorithm is implemented based on nano-vllm, which shows that the proposed method can be integrated into LLM serving engines and works well along with other parts in the serving engine. From this perspective, this paper has a profound contribution to the LLM inference community, and may potentially be adopted in multiple downstream application scenarios.

3. Detailed and sufficient empirical results show that the proposed method can significantly improve the task performance of both top-k based and top-p based methods. Efficiency gains are also notable under different settings.

## Weaknesses
---

1. The benchmark tasks seem to be too hard for KV cache compression. Take table 1 as an example, the full attention reference score of AIME24 is 79.9%, but the highest score of HARD-KV is 57.9%, where there is still a 22% performance degradation. One simple way to enhance the soundness of this paper is to include more suitable benchmarks, such as MATH-500 [1] and GAOKAO-Bench, which are also reasoning (long-generation) tasks.

2. There are multiple system-level optimizations proposed in Section 3.1, but there is no detailed ablation studies to show that each of the optimization is effective. Please provide an ablation study aiming at demonstrating the effective of the system-level design. If such experiment is not applicable, please provide an detailed explanation.

3. Although this paper aims at compressing the KV cache at decoding time, long-context prefill also introduce a large KV cache overhead, and the baseline SnapKV is also able to reduce the KV cache size right after each layer's prefill. From this perspective, long-context input tasks, such as LongBench [3] and RULER [4], should be taken into consideration. Math reasoning tasks cannot fully show the algorithm's performance on handling long-context retrieval task. Therefore, please include at least one long-context input benchmark.


---

[1] HuggingFaceH4/MATH-500

[2] Evaluating the Performance of Large Language Models on GAOKAO Benchmark

[3] LongBench: A Bilingual, Multitask Benchmark for Long Context Understanding

[4] RULER: What's the Real Context Size of Your Long-Context Language Models?

---

> ### Author Rebuttal · Authors · 2026-03-31
>
> ## R1 [W1] Evaluations on Long-generation tasks
> **Dataset Choice**
>
> We appreciate the suggestion to include additional benchmarks. As detailed in Appendix C.1, we prioritized the U-Math dataset over MATH500. While MATH500 is a standard benchmark, its relatively short average output lengths and potential overfitting on Qwen backbones make it less suitable. U-Math, featuring university-level reasoning and long-form generation, provides a more rigorous and realistic validation of our method's effectiveness.
>
> **Model Scaling**
>
> To further demonstrate robustness, we conducted experiments on Qwen3-32B (budget=4096, p=0.90). The results confirm that larger base models are significantly less prone to compression loss, with our method maintaining high performance even under aggressive budget constraints. This underscores the scalability of our approach for state-of-the-art, high-parameter models.
>
> | | aime24 | | aime25 | | u-math |  |
> |-|-|-|-|-|-|-|
> | | acc| avg. len.|acc|avg. len.|acc|avg. len. |
> |base|76.7|13104| 70.0 |16852|56.0|9169|
> |snapkv|53.3| 24862| 33.3| 24256|50.0|21097|
> |+ours|60.0|21549|46.7|22901|56.0|21778|
> |rkv|60.0|19911|40.0|21275|50.0|10537|
> |+ours|66.7|17030|50.0|21427|52.0|10123|
> |vanilla|40.0|25918|33.3|28701|48.0|22183|
> |+ours|50.0|22164 |43.3|24474|52.0|14711|
>
> Results on Qwen3-32B demonstrate that the performance gap between compressed and baseline models narrows as model scale increases. While baseline performance remains relatively consistent across different model sizes, the 32B model exhibits significantly higher resilience to KV eviction. This is further validated by near-lossless performance on the U-Math dataset, confirming that our method effectively preserves complex reasoning capabilities in larger architectures.
>
> We appreciate the suggestion to broaden our evaluation. While we would like to prioritize evaluations on GAOKAO and MATH500, we might add the comprehensive results in the next version of our paper. These additions will provide a more comprehensive and robust validation of our compression method across varying task complexities and output lengths.
>
> ## R2 [W2] System-level Abalations
>
> Our system-level ablations utilize unoptimized naive implementations as baselines, as no existing inference engines support dynamic headwise budgeting. We evaluate two distinct optimizations: Index Rewrite and Cascade Cache. The two methods operate one different levels: Index Rewrite on kernel-level, and Cascade Cache on engine-level.
>
> Index Rewrite (See Figure 5) delivers a 2x kernel-level speedup (27.08 ms vs. 53.26 ms) by improving computation efficiency; despite higher preparation overhead (8.14 ms vs. 1.14 ms) and temporal memory costs, it yields a net positive impact on the Memory-Time-Integral (MTI) metric (See discussion in Appendix B.2).
>
> Regarding end-to-end performance (Table 3), while Top-k pruning serves as a performance upper bound due to its native CUDA Graph compatibility, Top-p selection without Cascade Cache suffers from blocking latency and uneven budget allocation. Cascade Cache resolves these bottlenecks by enabling CUDA Graph support for headwise dynamic budgets, effectively pipelining the KV selection and decoding processes to maintain high system throughput.
>
> | Thr. (AIME 24, Qwen3-8B) | snapkv | vanilla |
> |-|-|-|
> |full-kv|-|346|
> |+ top-k (w/o. CUDA Graph) | - |470 |
> |+ top-k (w. CUDA Graph) | - | 806 |
> |+ top-p|301|279|
> |+ Cascade Cache (w/o. CUDA Grpah) |413|427|
> |+ Cascade Cache (w. CUDA Grpah)|632|732|
>
> ## R3 [W3] Evaluations on Long-context prefill tasks
>
> While our current system-level architecture is not primarily optimized for the long-context prefilling phase, the Top-p dynamic selection with Logits Calibration remains highly effective for traditional one-shot compression following prefilling.
>
> This post-prefilling application allows for significant KV cache reduction before the decoding phase begins. We validate the efficacy of this approach through extensive experiments on RULER and LongBench, demonstrating that our algorithmic framework maintains strong performance.
>
> ||base|snapkv|+ours|rkv|+ours|vanilla|+ours|
> |-|-|-|-|-|-|-|-|
> |RULER (32k, avg. on all tasks)|80.78|69.12|67.4|67.67|70.11|69.12|70.55|
> |LongBench (avg. on single doc QA) |28.02|28.27|28.71|27.67|29.10|26.84|28.40|
>
> ||base|snapkv |+ours|rkv| +ours | vanilla | +ours |
> |-|-|-|-|-|-|-|-|
> |Qasper|26.5|26.5|27.5|26.9|28.7|25.6|26.7|
> |NrtvQA|19.3|20.0|20.4|18.0|20.1|18.2|20.1|
> |MF-en|38.2|38.2|38.e|38.1|38.5|36.7|38.3|
>
> **Extension to Long-Context Prefilling**
>
> To support long-context scenarios, we envision integrating chunked-prefill, particularly for streaming applications like video LLM serving. In this pipeline, compression is triggered on a per-chunk basis and executed asynchronously within the Cascade Cache. This architecture enables efficient, sequential KV management without stalling the prefill stream, extending our dynamic selection benefits to the pre-decoding phase.

---

> > ### Author Rebuttal · Reviewer_5uEu · 2026-04-01
> >
> > All questions resolved. I'm keeping my positive score.

---

> > > ### Author Response · Authors · 2026-04-07
> > >
> > > Thank you very much for your encouraging feedback and for the time spent reviewing our manuscript. We are pleased that the core contributions of our work were well-received. We look forward to incorporating your suggestions—particularly regarding evaluations on broader benchmarks—to further strengthen the final version of our manuscript.
> > >
> > > Best regards, Authors of Submission 6322.

---

### Official Review · Reviewer_aEHh · 2026-03-15

**Soundness:** 3
**Presentation:** 3
**Significance:** 2
**Originality:** 3
**Overall Recommendation:** 3
**Confidence:** 4

**Summary:**

This paper proposes Hard-KV, which resolves the "static-dynamic" mismatch due to Top-p selection/sampling of tokens; more explicitly, Hard-KV addresses the challenges of PagedAttention fragmentation and CUDA graph incompatibility, based on the proposed cascade cache hierarchy and continuous compression pipeline. Experiments on math-reasoning benchmarks verify that Hard-KV achieves up to 2X throughput improvement over static baselines while maintaining high-quality generation. Please see my detailed comments below.

**Compliance With Llm Reviewing Policy:**

Affirmed.

**Key Questions For Authors:**

1. In my opinion, the speedup may come from system optimization such as CUDA graph compatibility, but not from better compression. How much of the gain (in speedup) comes from system optimization vs. compression algorithm?
2. Only Qwen3-8B and Qwen3-4B (in the appendix) were used for experimentation and evaluation. Does Hard-KV scale to ≥30B models?
3. Please clarify my concern/confusion as indicated in "Weaknesses" (Item 3).

**Limitations:**

Yes.

**Strengths And Weaknesses:**

Strengths
1. Addressing the challenges of PagedAttention fragmentation and CUDA graph incompatibility is practical (but the results do not seem good).
2. The proposed framework based on cascade cache hierarchy and continuous compression pipeline is interesting and conceptually useful.
3. The integration of Hard-KV into Nano-vLLM is valuable, and may enhance the visibility of this work.

Weaknesses
1. Limited novelty at the algorithmic level: most components are adaptations of existing ideas, e.g., Top-k/Top-p selection, token eviction, head-adaptive allocation.
2. Only Qwen3-8B and Qwen3-4B (in the appendix) were used for experimentation and evaluation.
3. Results are not good. Although Hard-KV improves SnapKV and RKV significantly, the improved accuracy is still far from the reference. For example, in Table 1, when the budget size is 2048, Hard-KV improves SnapKV from 21.3% to 40.2%, but the improved accuracy of 40.2% is still far from 79.9% (reference); when the budget size is 4096, Hard-KV fails to improve SnapKV!?
4. The design of Table 3 needs to be improved. It took me a whole to understand what each number in the lower half of the table means.

---

> ### Author Rebuttal · Authors · 2026-03-31
>
> ## R1 [W1] Novelty
> Our work bridges the gap between theoretical speedups and practical performance. While dynamic budget allocation (e.g., Top-p) improves attention accuracy, existing methods struggle to achieve real-world speedups. We provide two primary novel contributions:
>
> **Native Top-p Selection**
> Unlike Twilight [1], which relies on a two-stage "Top-K then Top-p" refinement, our method features native Top-p selection. We introduce Logits Calibration, which maps transformed logits to the same density as raw logits. This provides a unified framework that preserves intrinsic attention properties while allowing for flexible designs (see Appendix G).
>
> **Sequence-level hybrid of KV Cache memory**
> Translating head-wise dynamic sparsity into hardware speedups is non-trivial. Cascade Cache enables a sequence-level hybrid of constant- and linear-sized memory. This architecture:
> - Resolves the conflict between dynamic head-wise budgets and static system constraints.
> - Enables conversion between different cache areas.
> - Supports streaming pre-filling and online KV compression (e.g., for long-form video understanding).
>
> In summary, beyond algorithmic efficiency, our work provides the system-level foundation necessary to make dynamic KV eviction practical and compatible with modern inference optimizations.
>
> [1] Twilight: Adaptive Attention Sparsity with Hierarchical Top-p Pruning (NeurIPS 2025)
>
> ## R2 [Q1 W4] Ablation in Speedups
> Cascade Cache is designed for decoding-time compression (as also explained in Response to 5CZK R1), which improves the efficiency from optimized compression execution along with compatibility of CUDA Graphs.
>
> Here are the detailed breakdowns of e2e throughput (tokens/s) gain that show the system-level ablation.
>
> |Thr. (AIME 24, Qwen3-8B)|SnapKV|vanilla|
> |-|-|-|
> |full-kv |-| 346 |
> |+ top-k (w/o. CUDA Graph)| -|470|
> |+ top-k (w. CUDA Graph)|-|806|
> |+ top-p  | 301| 279|
> |+ Cascade Cache (w/o. CUDA Grpah) |413|427|
> |+ Cascade Cache (w. CUDA Grpah)|632|732|
>
> As shown in Table 3, vanilla Top-p can degrade system throughput due to selection overhead and non-contiguous memory layouts. However, our method bridges this gap, enabling Top-p to be comparable with natively CUDAGraph-compatible Top-k methods.
>
> We appreciate the suggestion regarding Table 3 and will refine it to more clearly demonstrate these e2e performance gains.
>
> ## R3 [W2 Q2] Scale to Larger Models
> Scaling to 30B+ models is a significant system challenge; on a single GPU, these models leave <20GB for KV cache, making multi-GPU parallelism essential. We employ Tensor Parallelism (TP) rather than Pipeline Parallelism (to avoid bubbles) or Expert Parallelism (due to CUDAGraph incompatibility).
>
> Because TP naturally partitions attention heads, each instance performs head-wise KV eviction independently. To minimize communication overhead, our customized nanovllm engine implements two key optimizations:
> - Asynchronous Mask Management: We avoid synchronizing head-wise sparse masks by managing them on the CPU, preventing frequent CPU-GPU synchronization stalls.
> - Localized Metadata: Sequence-level metadata (e.g., head-wise block tables) is stored locally within each instance, eliminating the need for cross-instance serialization or communication.
>
> With the scale-out system (code will be updated), we validated the performance on Qwen3-32B.
>
> |Acc. |base|snapkv|+ours|rkv|+ours|vanilla|+ours|
> |-|-|-|-|-|-|-|-|
> |aime24|76.7|53.3|60.0|60.0|66.7|40.0|50.0|
> |u-math|56.0|50.0|56.0|50.0|52.0|48.0|52.0|
>
> As shown in the results, Qwen3-32B has more moderate drops in performance compared to smaller counterparts, demonstrating the effectiveness of our methods to scale.
>
> ## R4 [W3 Q3] Loss in accuracy
> **The Challenge of Decoding-time Compression**
> Decoding-time KV compression is inherently more difficult, since previous compression can affect later ones. As a heuristic, FP8 quantization (50% compression) sits at the boundary of lossless performance (and also struggles at reasoning tasks), our experiments use a 4k token budget for 12k–16k sequences. Although a higher budget can bring better performance, we keep the setting to better compare across different methods.
>
> **Further efforts to validate**
> Despite the over-complex reasoning tasks, our method maintains high accuracy across the following benchmarks:
>
> - University-level Math (u-math): Our method maintains the integrity of complex, multi-stage logic.
> - Retrieval Tasks: We achieve near-lossless accuracy, with our method occasionally outperforming the baseline by filtering out non-essential "noise" tokens. (See Response to Reviewer 5CZK R1)
>
> While predicting exact performance degradation remains a complex challenge, we provide hyperparameter tuning insights in Section 4.2 to mitigate these effects. Although future work may yield even more performance-preserving algorithms, our work offers significant system-level value by taming dynamic KV memory.

---

> > ### Author Rebuttal · Reviewer_aEHh · 2026-04-05
> >
> > Thanks to the authors for the rebuttal. I would keep my score of 3 (weak reject) due to the concern on novelty.

---

> > > ### Author Response · Authors · 2026-04-07
> > >
> > > We sincerely thank you for your recognition, and are delighted that our rebuttal fully addressed your questions. While we respect the concerns regarding novelty, we remain confident that our contributions are vital for the emerging task of decoding-time KV compression. Furthermore, your suggestions on scaling have helped us better demonstrate the effectiveness of our methods. As noted in your review, our implementation on an open-source inference engine, now further extended with tensor parallelism, represents a significant contribution and will be made publicly available. We thank you again for your engagement and look forward to integrating your feedback to improve the overall quality of the next version.
> > >
> > > Best regards, Authors of Submisson 6322.

---

### Official Review · Reviewer_5CZK · 2026-03-22

**Soundness:** 3
**Presentation:** 2
**Significance:** 2
**Originality:** 3
**Overall Recommendation:** 4
**Confidence:** 3

**Summary:**

This paper proposes HARD-KV, a framework for decoding-time KV-cache compression that aims to reconcile head-adaptive token selection with the static memory/layout requirements of modern inference systems. The proposed system consists of a three-tier cascade cache hierarchy, a logits calibration mechanism to prevent selection collapse, and an index regularization engine to rewrite fragmented sparse indices into a regularized, contiguous layout, restoring compatibility with CUDA graphs. Experiments on long math-reasoning benchmarks show improved throughput and, in several settings, better accuracy-efficiency tradeoffs than adapted baselines.

**Compliance With Llm Reviewing Policy:**

Affirmed.

**Final Justification:**

My concern has been fully addressed. I will maintain my positive score.

**Key Questions For Authors:**

1. Can the authors provide a cleaner ablation separating the effects of calibration, Cascade Cache, and index rewrite?
2. How does HARD-KV perform on non-math long-context tasks, especially retrieval-style benchmarks or long-document QA?
3. In Table-3, what do "SR" and "CR" denote? It is also unclear why different metrics are used across datasets.
4. Could the authors report the compression ratio (i.e., KV-cache memory savings) and the computation-saving ratio (e.g., attention FLOPs reduction) to give a clearer picture of where the improvement comes from?
5. In Table-3, why does Top-p = 0.60 sometimes achieve higher accuracy than Top-p = 0.90? Some discussion of this non-monotonic trend would be helpful.

**Limitations:**

Yes

**Strengths And Weaknesses:**

**Strength:**
* The paper addresses an important practical problem by bridging the gap between dynamic KV-cache compression methods and the constraints of real-world inference systems.
* The calibration idea is interesting, mapping heterogeneous selectors into a common probabilistic budget space enables more principled comparison and deployment.
* The reported throughput improvements are promising.

**Weakness:**
* The evaluation is heavily focused on math reasoning. Results on broader long-context benchmarks would be helpful to confirm the generalizability of the proposed method.
* There is a lack of ablation studies isolating the effect of the calibration mechanism from the cascade cache and index rewrite design.
* The remaining accuracy gap relative to full attention is still fairly large.

---

> ### Author Rebuttal · Authors · 2026-03-31
>
> ## R1 [W1, Q2] Evaluations on general Long-Context benchmarks
> **Decoding-time KV compression**
>
> HARD-KV focuses on decoding-time compression, specifically addressing reasoning tasks (e.g., coding, math) where short inputs and massive outputs necessitate **continuous compression**.
>
> Unlike traditional long-context models that compress once after prefill, continuous compression introduces two critical bottlenecks:
>
> - Memory Inefficiency: Random or head-wise eviction disrupts block contiguity, reducing memory fabric utilization and increasing kernel-level latency.
> - Execution Latency: Standard compression is "blocking," which halts normal decoding steps.
>
> By  **Cascade Cache**, HARD-KV can reduce memory fabrics and minimize latency with pipelining compression and decoding.
>
> **General long-context tasks**
> While HARD-KV’s pipelining advantages are evident during decoding-time compression, the headwise Top-p selection remains effective for one-shot compression after the prefill phase.
>
> We evaluated our method using the RULER and LongBench benchmarks.
> (Context length for RULER set as 32k, Qwen3-32B, 4096 budget).
>
> | |base|snapkv|+ours|rkv|+ours|vanilla|+ours|
> |-|-|-|-|-|-|-|-|
> |RULER (avg. )|80.8|69.1|67.4|67.7|70.1|69.1|70.6|
> |LongBench (avg. on single doc QA) |28.0|28.2|28.7|27.6|29.1|26.8|28.4|
>
> While highly effective for retrieval tasks, our Top-p budget allocation, integrated with logits calibration, is specifically suitable for decoding-time compression.
>
> - Adaptive Efficiency: Headwise dynamic allocation utilizes "attention density" to eliminate redundant KV storage.
>
> - "Lazy" Eviction: While Top-K always fills a fixed budget , Top-P selection only allocates space as needed. This efficiency enables a "cautious" or "lazy" eviction strategy for tokens appended afterwards.
>
> For generalized long-input scenarios, HARD-KV can integrate with chunked or streaming prefill. By periodically processing chunks, our pipeline applies these efficiency gains to traditional long-context workflows, maintaining performance across varying workload types.
>
> ## R2 [W2, Q1, Q4] Detailed Ablations
> We categorize our ablations into two parts: algorithm-level and system-level.
>
> **Algorithm-level**
> We identify **logits-calibration** as the critical algorithm that solve the Top-p divergence.
>
> As illustrated in Figure 2, at a fixed budget of p=0.90, uncalibrated SnapKV-transformed logits suffer a drastic drop in selection density, falling from $e^7\approx 1000$ to $e^3 \approx 10$ tokens compared to raw attention. By mapping the distribution shape back to the density of raw attention, our calibration method preserves the number of selections under a unified Top-p budget, ensuring both efficiency and algorithmic consistency across different methods.
>
> We make a quick proof-of-concept experiment for SnapKV without calibration on AIME24 (Qwen3-8B).
>
> | p=0.90, budget=4096 | AIME24 |
> | - | - |
> | snapkv | 3.33  |
> | +logits calibration | 56.7  |
>
> **System-level**
>
> Our system-level ablations utilize unoptimized naive implementations as baselines, as no existing inference engines support dynamic headwise budgeting. We evaluate two distinct optimizations: Index Rewrite and Cascade Cache.
>
> Index Rewrite (See Figure 5) delivers a 2x kernel-level speedup (27.08 ms vs. 53.26 ms) by improving computation efficiency; despite higher preparation overhead (8.14 ms vs. 1.14 ms) and temporal memory costs (See Appendix B.2).
>
> Regarding e2e performance (Table 3), Top-k pruning serves as a performance upper bound due to its native CUDA Graph compatibility. Cascade Cache resolves bottlenecks mentioned above and enables CUDA Graph support for headwise dynamic budgets.
>
> | Thr. (AIME 24, Qwen3-8B) | snapkv | vanilla |
> | - | - | - |
> | full-kv | - | 346  |
> | + top-k (w/o. CUDA Graph)  | - | 470 |
> | + top-k (w. CUDA Graph)  | - | 806 |
> | + top-p  | 301    | 279     |
> | + Cascade Cache (w/o. CUDA Graph) | 413 | 427 |
> | + Cascade Cache (w. CUDA Graph)   | 632 | 732 |
>
> ## R3 [Q3] (typos) SR and CR
>
> Sorry for the confusion caused by this typo, here should be SU (Sparsity Utilization, instead of SR or CR). We define SU as $ \frac{Overall\ Sparsity}{Max\ Sparsity} $ representing the ratio of the "taking-effect" KV cache to the total allocated budget. As visualized by the volume of the colored bars within the cube in Figure 9, this metric provides a precise measure of memory efficiency by quantifying the proportion of the allocated KV cache that is actively utilized during inference.
>
> ## R4 [Q5] Non-monotonic trend of different sparsity ratio
>
> The real sparsity ratio is strictly governed by the Top-K constraint, whereas the Top-P budget acts as a modulator for the growth rate rather than a hard threshold (see Figure 8, and this is correlated to the **"lazy" eviction** as mentioned above). This distinction explains the non-monotonic trends observed relative to the Top-P budget, as it dictates the accumulation rate of the cache rather than enforcing a fixed capacity limit.

---

> > ### Author Rebuttal · Reviewer_5CZK · 2026-04-03
> >
> > Thank you for the response. I tend to keep my score.

---

> > > ### Author Response · Authors · 2026-04-07
> > >
> > > We sincerely thank you for the positive feedback. We hope our clarifications regarding the unique challenges of decoding-time KV compression have effectively addressed your concerns and highlighted the significance of our contributions. Your insightful questions are vital to improving the next version of our paper, and we will ensure they are fully integrated.
> > >
> > > Best regards, Authors of Submisson 6322.

---

### Decision · Program_Chairs · 2026-04-30

**Decision:**

Accept (regular)

**Comment:**

This submission proposes a head-adaptive regularization framework for decoding-time KV compression that bridges the gap between dynamic compression methods and the static memory requirements of practical inference engines. The reviewers found the problem important and the system contribution meaningful, particularly the integration with nano-vLLM and the practical handling of issues such as fragmentation and CUDA graph compatibility. While concerns were raised about the algorithmic novelty, evaluation scope, and the remaining accuracy gap relative to full attention, the authors’ response addressed most of these issues. I therefore recommend acceptance.

I also note a bibliographic error in the reference list (e.g., the citation for the Adam optimizer), which should be corrected in the final version.